# CAN EUCLIDEAN SYMMETRY HELP IN REINFORCEMENT LEARNING AND PLANNING?

## ABSTRACT

In robotic tasks, changes of reference frames do not affect the underlying physics of the problem. Isometric transformations, including translations, rotations, and reflections, collectively form the Euclidean group. In this work, we study reinforcement learning and planning tasks that have Euclidean group symmetry. We show that MDPs with continuous symmetries have linear approximations that satisfy steerable kernel constraints, which are widely studied in equivariant machine learning. Guided by our theory, we propose an equivariant model-based RL algorithm algorithm, which is based on sampling-based MPPI for continuous action spaces. We test our proposed equivariant TD-MPC algorithm on a set of standard RL benchmark tasks. Our work shows that equivariant methods can give a great boost in performance on control tasks with continuous symmetry.

## 1 INTRODUCTION

Robot decision-making tasks often involve the movement of robots in two or three-dimensional Euclidean space. Different reference frames can be used to model the robot and environment, but the *underlying physics* of the system must be *independent* of the choice of reference frame (Einstein, 1905). The set of all such reference frame transformations is called the Euclidean group $E(d)$. In this work, we show that utilizing the Euclidean frame symmetry inherent in many robotic planning and control problems allows for the design of more efficient learning algorithms. The use of symmetry in decision-making has been studied in model-free and model-based reinforcement learning (RL), planning, optimal control, and other related fields (Ravindran & Barto, 2004; Zinkevich & Balch, 2001; van der Pol et al., 2020a; Mondal et al., 2020; Wang et al., 2021; Zhao et al., 2022b). Despite this, there is no unified theory of how symmetry can be utilized to develop better RL or planning algorithms for robotics applications.

In many problems in robotics, we are interested in the Markov Decision Process (MDP) that describes a robot moving in 2D or 3D space. Motivated by the study of *geometric graphs* and geometric deep learning (Bronstein et al., 2021), we define *Geometric MDPs* as the class of MDPs that correspond to the decision process of a robot moving in Euclidean space. The question that we aim to answer is: *Can Euclidean symmetry guarantee benefits in (model-based) RL algorithms?* To answer it, we aim to first formally describe what "benefits" mean and how symmetry enables them, then show a model-based RL algorithm that is developed with the guidance of the theory.

To begin, we present a theoretical framework that studies the linearized dynamics of geometric MDPs and shows that the matrices that appear in linearized dynamics are $G$-steerable kernels (Cohen & Welling, 2016d). Using recent results on parameterizations of steerable kernels (Lang & Weiler, 2020b), we show that the steerable kernel solution significantly reduces the number of parameters needed to specify the linearized dynamics. We can use it to predict parameter reduction for tasks with geometric structure. The reduction is infinite for continuous tasks with continuous symmetry, such as moving 2D particle.

Inspired by the theoretical results which show that equivariant versions of linearized model-based approaches contain a smaller number of parameters than general models, we propose an equivariant sampling-based model-based RL algorithm for Geometric MDPs. It is based on Model Predictive Path Integral (MPPI); we propose strategy that enforces symmetry on the sampling process: if the input state is rotated, the output action should be rotated accordingly, as demonstrated in Figure 1.

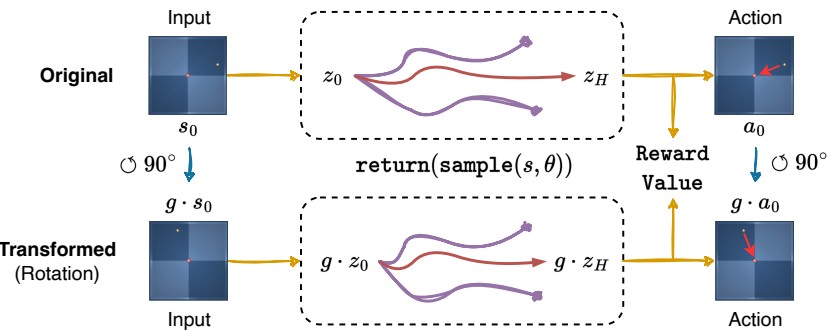

Figure 1: Illustration of equivariance in the proposed sampling-based planning algorithm $a_0 = \texttt{plan}(s_0)$: if the input state is rotated, the output action should be rotated accordingly. This requires the learned functions are $G$-equivariant or $G$-invariant networks and a special sampling strategy, introduced in our method.

Our method extends the prior work from (1) planning on 2D grids with value-based planning (Zhao et al., 2022b) and (2) model-free equivariant RL (van der Pol et al., 2020b; Wang et al., 2021) to continuous state and action spaces. We take inspiration from geometric deep learning (Bronstein et al., 2021) and consider the features in neural networks to transform under Euclidean symmetry. Our algorithm is constructed to be equivariant with respect to changes of the reference frame, which is usually known beforehand. We evaluate the proposed algorithm on DeepMind Control suite and MetaWorld continuous-control tasks and show its sample efficiency against non-equivariant methods, which demonstrates the benefits of equivariance in model-based RL with sampling-based planning (MPPI) and the value of our theory.

Our contributions can be summarized as follows: *(i)* We define a class of MDPs that correspond to the movement of a physical agent in two or three dimensional Euclidean space ("Geometric MDPs"). *(ii)* By analyzing the linearization of Geometric MDPs, our theory shows a reduction in the number of free parameters in the ground-truth linearized dynamics and optimal control policy. *(iii)* Motivated by our theory, we propose a sampling-based model-based RL algorithm that leverages Euclidean symmetry for Geometric MDPs. *(iv)* Our empirical results demonstrate the effectiveness of our method in solving MDPs on control tasks with continuous symmetries.

## 2 PROBLEM STATEMENT: SYMMETRY AND CHOICE OF REFERENCE FRAME

To theoretically study how symmetry benefits in solving MDPs, we describe the source of symmetry and define a class of MDPs that has symmetry constraints and can be linearized.

### 2.1 GEOMETRIC STRUCTURE IN MDP

The set of all isometric changes of reference frame form the Euclidean symmetry group $\mathrm{E}(d)$ (Bronstein et al., 2021; Weiler & Cesa, 2021; Lang & Weiler, 2020b). Any subgroup of $\mathrm{E}(d)$ can be expressed in semi-direct product form as $\left(\mathbb{R}^d, +\right) \rtimes G$, where $G$ is the stabilizer group of origin and the action on a vector $x$ includes a translation part $t$ and rotation/reflection part $g$, i.e., $x \mapsto (tg) \cdot x := gx + t$ (Lang & Weiler, 2020b).

To transform an MDP (to a different reference frame), we require the MDP to have the group $G$-action on both the state and action space (Zhao et al., 2022b; Wang et al., 2021; van der Pol et al., 2020b). This definition unifies different types of prior work and allows the state and actions spaces to be a any spaces equipped with a $G$-action (van der Pol et al., 2020b; Wang et al., 2021; Zhao et al., 2022b; Teng et al., 2023). The compact group $G \leq \mathrm{GL}(d)$ can be any group, including the group of proper 3D transformations $\mathrm{SO}(3)$ or finite subgroups like the icosahedral group or cyclic groups (Brandstetter et al., 2021). We define a class of MDPs with geometric structure, extending a previously studied discrete case (Zhao et al., 2022b).

**Definition 1 (Geometric MDP)** *A Geometric MDP (GMDP) $\mathcal{M}$ is an MDP with a (compact) symmetry group $G \leq \mathrm{GL}(d)$ that acts on the state and action space. It is written as a tuple*

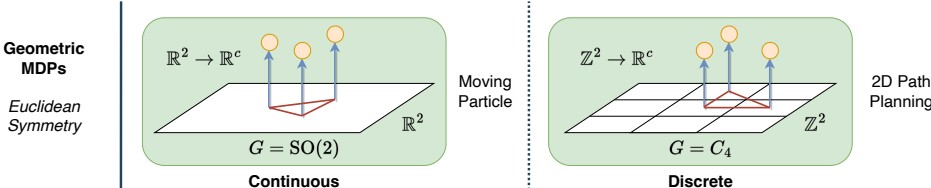

Figure 2: Illustration of MDPs with underlying geometric structure, e.g., a 2D particle moving or a path planing problem, which have 2D rotation groups $G$ that have $G$-action on the MDP state and action space.

$\langle \mathcal{S}, \mathcal{A}, P, R, \gamma, G, \rho_{\mathcal{S}}, \rho_{\mathcal{A}} \rangle$. *The state and action spaces $\mathcal{S}, \mathcal{A}$ have (continuous) group actions that transform them, defined by $\rho_{\mathcal{S}}$ and $\rho_{\mathcal{A}}$.*

The *symmetry properties* in MDPs are specified by equivariance and invariance of the transition and reward functions respectively (Zinkevich & Balch, 2001; Ravindran & Barto, 2004; van der Pol et al., 2020a; Wang et al., 2021; Zhao et al., 2022b; 2023a):

$$\forall g \in G, \forall s, a, s', \qquad P(s' \mid s, a) = P(g \cdot s' \mid g \cdot s, g \cdot a) \tag{1}$$

$$\forall g \in G, \forall s, a, \qquad R(s, a) = R(g \cdot s, g \cdot a) \tag{2}$$

where $g$ acts on the state and action spaces by group representations $\rho_{\mathcal{S}}$ and $\rho_{\mathcal{A}}$ respectively. For example, the *standard representation* $\rho_{\text{std}}(g)$ of $\text{SO}(2)$ assigns each rotation $g \in \text{SO}(2)$ a 2D rotation matrix $R_{2 \times 2}$, while the *trivial representation* $\rho_{\text{tri}}(g)$ assigns identity $\mathbf{1}_{1 \times 1}$ to all $g$.

**Continuous $G$-action.** Compared to prior work, we additionally require *continuous* group action $\cdot_G : G \times X \to X$ and find it gives promising theoretical results, which is optional for implementation. If the $G$-actions on $\mathcal{S}$ and $\mathcal{A}$ are continuous[1], there is an interesting geometric interpretation based on fiber bundle theory (Husemöller, 2013). The linearized dynamics of a system are much more constrained[2] (i.e., have fewer free parameters) when the system has continuous $G$-action. See Appendix D for more detail.

**Examples.** We list some examples in the Table 1 to demonstrate that the definition covers previous work (a homogeneous space, a group, or any other space as long as equipped with a $G$-action (van der Pol et al., 2020b; Wang et al., 2021; Zhao et al., 2022b; Teng et al., 2023)) and what symmetry could bring. We list the symmetry group $G$, the state and action spaces $\mathcal{S}, \mathcal{A}$, and provide how symmetry $G$ reduces the space via quotient $\mathcal{S}/G$ and how "large" a group $G$ can relate different states in that MDP via orbit $Gx$. We provide detailed explanation in Appendix B.4.

## 2.2 RELATED TOPICS

We discuss geometric graphs and the use of symmetry in reinforcement learning. For further discussion, please see Appendix B.

**Geometric graphs.** Our definition of GMDP is closely related to the concept of *geometric graphs* (Bronstein et al., 2021; Brandstetter et al., 2021), which model MDPs as state-action connectivity graphs. Previous works studied algorithmic alignments and dynamic programming on geometric graphs (Xu et al., 2019; Dudzik & Veličković, 2022). We propose extending this concept to include additional geometric structures by embedding the MDP into a geometric space such as $\mathbb{R}^2$ or $\mathbb{R}^3$. In discussion of GMDP, we focus on 2D and 3D Euclidean symmetry (Lang & Weiler, 2020b; Brandstetter et al., 2021; Weiler & Cesa, 2021), with the corresponding symmetry groups of $\text{E}(2)$ and $\text{E}(3)$, respectively. The relation between equivariant message passing and dynamic programming / value iteration on geometric MDPs is discussed in Section 3.

**Symmetry in MDPs.** Symmetry in decision-making tasks has been explored in previous works on MDPs and control, with research on symmetry in MDPs with no function approximation (Ravindran & Barto, 2004; Ravindran & Barto; Zinkevich & Balch, 2001) and symmetry in model-free (deep)

---

[1]If the group $G$ is additionally a *compact* Lie group, there exists a map $p : \mathcal{S} \times \mathcal{A} \mapsto \mathcal{B}$ that projects the state-action space $\mathcal{S} \times \mathcal{A}$ to a lower dimensional base space $\mathcal{B}$ (Cohen et al., 2020b). The existence and smoothness of the projection $p$ can be established using principal bundle theory.

[2]Continuous symmetries correspond to conservation laws, while discrete (non-differentiable) symmetries do not have corresponding conservation laws (Zee, 2016).

Table 1: Examples of geometric MDPs. $G$ denotes the MDP symmetry group. $\mathcal{S}$ denotes the MDP state space. $\mathcal{A}$ denotes the MDP action space. We can quantitatively measure the savings obtained by exploiting equivariance. "Images" refers to panoramic egocentric images $\mathbb{Z}^2 \to \mathbb{R}^{H \times W \times 3}$. $\circ$ denotes group element composition. We list the quotient space $\mathcal{S}/G$ to give intuition on savings. The $Gx = \{g \cdot x \mid g \in G\}$ column shows the $G$-orbit space of $\mathcal{S}$ ( $\cong$ denotes isomorphic equivalence).

| ID | $G$ | $\mathcal{S}$ | $\mathcal{A}$ | $\mathcal{S}/G$ | $Gx$ | Task |
|----|-----|-----|-----|-----|-----|------|
| 1 | $C_4$ | $\mathbb{Z}^2$ | $C_4$ | $\mathbb{Z}^2/C_4$ | $C_4$ | 2D Path Planning (Tamar et al., 2016) |
| 2 | $C_4$ | Images | $C_4$ | $\mathbb{Z}^2/C_4$ | $C_4$ | 2D Visual Navigation (Zhao et al., 2022b) |
| 3 | SO(2) | $\mathbb{R}^2$ | $\mathbb{R}^2$ | $\mathbb{R}^+$ | $S^1$ | 2D Continuous Navigation |
| 4 | SO(3) | $\mathbb{R}^3 \times \mathbb{R}^3$ | $\mathbb{R}^3$ | $\mathbb{R}^+ \times \mathbb{R}^3$ | $S^2$ | 3D Free particle (with velocity) |
| 5 | SO(3) | $\mathbb{R}^3 \rtimes \text{SO}(3)$ | $\mathbb{R}^3 \times \mathbb{R}^3$ | $\mathbb{R}^+ \times \mathbb{R}^3$ | $S^2$ | Moving 3D Rigid Body |
| 6 | SO(2) | SO(2) | $\mathbb{R}^2$ | $\{e\}$ | $S^1$ | Free Particle on SO(2) $\cong S^1$ manifold |
| 7 | SO(3) | SO(3) | $\mathbb{R}^3$ | $\{e\}$ | $S^2$ | Free Particle on SO(3) (Teng et al., 2023) |
| 8 | SO(2) | SE(2) | SE(2) | $\mathbb{R}^2$ | $S^1$ | Top-down grasping (Zhu et al., 2022) |
| 9 | SO(2) | $(S^1)^2 \times (\mathbb{R}^2)^2$ | $\mathbb{R}^2$ | $S^1 \times (\mathbb{R}^2)^2$ | $S^1$ | Two-arm manipulation (Tassa et al., 2018) |

RL using equivariant policy networks (van der Pol et al., 2020a; Mondal et al., 2020; Wang et al., 2021). Additionally, the use of symmetry in value-based planning on a 2D grid is analyzed by Zhao et al. (2022b). We extend this line of work by focusing on MDPs with continuous state and action spaces and sampling-based planning/control algorithms.

## 3    THEORY: WHY IS SYMMETRY USEFUL IN GEOMETRIC MDPs?

The goal of this section is to provide theoretical guidance on assessing the potential benefits of symmetry in a Geometric MDP for a Reinforcement Learning (RL) algorithm, particularly when planning using learned dynamic models.

### 3.1    PROPERTIES OF GEOMETRIC MDPs

In RL, the optimal policy mapping is $G$-equivariant (Ravindran & Barto, 2004). To incorporate symmetry constraints, a strategy is to constrain the entire policy mapping to be equivariant: $a_t = \texttt{policy}(s_t)$ (van der Pol et al., 2020b; Wang et al., 2021; Zhao et al., 2022b; 2023a), as shown in Figure 1. Many model-based RL algorithms rely on iteratively applying Bellman operations (Sutton & Barto, 2018). Thus, we first show that symmetry $G$ in a Geometric MDP (GMDP) results in $G$-equivariant Bellman operator, which indicates that we can constrain the iterative process in model-based RL algorithms to be $G$-equivariant to exploit symmetry. Additionally, for GMDPs[3], a specific instance of DP-based algorithm, value iteration, can be connected with geometric graph neural network (Bronstein et al., 2021). These properties do not require linearization and do not require continuous group actions.

**Theorem 1** *The Bellman operator of a GMDP is equivariant under Euclidean group* $\text{E}(d)$.

**Theorem 2** *For a GMDP, value iteration resembles* $\text{E}(d)$*-equivariant geometric message passing.*

We provide proofs and derivation in Appendix D. This is an extension to the theorems in (Zhao et al., 2022b) on 2D discrete groups, where they showed that value iteration is equivariant under discrete subgroups of the Euclidean group: discrete translations, rotations, and reflections. We generalize this result to groups of the form of $(\mathbb{R}^d, +) \rtimes G$, where $G$ is continuous[4].

### 3.2    LINEARIZING GEOMETRIC MDPs: $G$-STEERABLE KERNEL CONSTRAINTS

---

[3]For non-geometric graphs, Dudzik & Veličković (2022) show the equivalence between dynamic programming on a (general non-geometric) MDP and a message-passing GNN.

[4]For the translation part, one may use relative/normalized positions or induced representations (Cohen et al., 2020b; Lang & Weiler, 2020b).

The dynamics function in GMDPs is generally nonlinear. In this subsection[5], we derive the iterative linearization of dynamics of GMDPs to get $G$-equivariant linear maps. We focus on the linearization for two reasons: *(1)* if infinitesimal group actions on state-action space exists, the symmetry of the nonlinear GMDP is *equivalent* to $G$-steerable constraints of the linear dynamics, *(2)* the linearized dynamics is connected to LQR and is easier to analyze, such as the dimensions of the (linear) dynamics function, policy function, and more.

**Iterative Linearization.** We assume the dynamics is deterministic $f : \mathcal{S} \times \mathcal{A} \to \mathcal{S}$ and *iteratively linearize* $f$ at each step. It is naturally connected to *time-varying* iterative Linear Quadratic Regulator (iLQR). We highlight the linearization procedure of $f(\boldsymbol{s}_t, \boldsymbol{a}_t)$, where matrices $A$ and $B$ depend *arbitrarily* on time step $t$. Later, we assume that it only depends on state and action $(\boldsymbol{s}_t, \boldsymbol{a}_t)$.

$$\text{Original:} \quad \boldsymbol{s}_{t+1} = f(\boldsymbol{s}_t, \boldsymbol{a}_t) \quad \to \quad \text{Linearized at step } t: \quad \boldsymbol{s}_{t+1} = A_t \cdot \boldsymbol{s}_t + B_t \cdot \boldsymbol{a}_t \quad (3)$$

**Theorem 3** *If a Geometric MDP has an infinitesimal $G$-action on the state-action space $\mathcal{S} \times \mathcal{A}$, the linearized dynamics is also $G$-equivariant: the matrix-valued functions $A : \mathcal{S} \times \mathcal{A} \to \mathbb{R}^{d_\mathcal{S} \times d_\mathcal{S}}$ and $B : \mathcal{S} \times \mathcal{A} \to \mathbb{R}^{d_\mathcal{S} \times d_\mathcal{A}}$ satisfy $G$-steerable kernel constraints.*

Under infinitesimal symmetry transformation $g \approx 1_G \in G$, the state and action spaces transform as $\boldsymbol{s} \mapsto \rho_\mathcal{S}(g) \cdot \boldsymbol{s}, \boldsymbol{a} \mapsto \rho_\mathcal{A}(g) \cdot \boldsymbol{a}$ where $\rho_\mathcal{S}$ and $\rho_\mathcal{A}$ are representations of the group $G$. Additionally, the dynamics must satisfy,

$$\rho_\mathcal{S}(g) \cdot f(\boldsymbol{s}, \boldsymbol{a}) = f(\rho_\mathcal{S}(g) \cdot \boldsymbol{s}, \rho_\mathcal{A}(g) \cdot \boldsymbol{a}) \quad (4)$$

Let us consider the linearized problem at point $p = (\boldsymbol{s}_0, \boldsymbol{a}_0)$. Assuming that the state and control do not change too drastically over a short period of time and that the time-varying $A$ and $B$ only depend on the linearization point $p$ but not other factors, we can approximate the true dynamics as

$$\boldsymbol{s}_{t+1} = A(p) \cdot \boldsymbol{s}_t + B(p) \cdot \boldsymbol{a}_t, \quad A : \mathcal{S} \times \mathcal{A} \to \mathbb{R}^{d_\mathcal{S} \times d_\mathcal{S}}, \quad B : \mathcal{S} \times \mathcal{A} \to \mathbb{R}^{d_\mathcal{S} \times d_\mathcal{A}}. \quad (5)$$

Now, linearizing $f$ and using the symmetry constraint, the matrix-valued functions $A(p)$ and $B(p)$ must satisfy the constraints

$$\forall g \in G, \quad A(g \cdot p) = \rho_\mathcal{S}(g) A(p) \rho_\mathcal{S}(g^{-1}), \quad B(g \cdot p) = \rho_\mathcal{S}(g) B(p) \rho_\mathcal{A}(g^{-1}) \quad (6)$$

so that $A$ is a $G$-steerable kernel with input and output representation $\rho_\mathcal{S}$, and $B$ has input type $\rho_\mathcal{S}$ and output type $\rho_\mathcal{A}$. The kernel constraints relate $A(p)$ and $B(p)$ at different points. We use the Figure 3 to demonstrate the idea. On each orbit (left), the constraints can be solved exactly: the matrices on same orbits (same colors) are related and have explicit parameterization given in Lang & Weiler (2020a). Thus, these matrices can be spanned on a basis (denoted by $K$) and live in a smaller "base" space $\mathcal{B} = X/G$ with a certain form $A_\downarrow : \mathcal{B} \to \mathbb{R}^{2 \times 2}$.

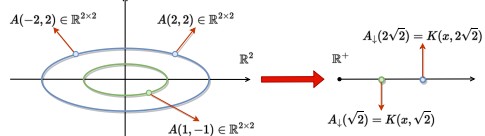

Figure 3: A schematic showing how a matrix-valued kernel $A : X \to \mathbb{R}^{2 \times 2}$ is constrained by the SO(2)-steerable kernel constraints on a set of orbits $A(g \cdot p) = \rho_{\text{out}}(g) A(p) \rho_{\text{in}}(g^{-1})$. This example is further explained in Appendix D.

**Benefits for control.** Symmetry further enables better control. We can further show that the policy and value function can be parameterized with fewer parameters based on the discrete algebraic Riccati equation (DARE) for time-varying LQR problem.

**Theorem 4** *The LQR feedback matrix in $a_t^\star = -K(p)s_t$ and value matrix in $V = s_t^\top P(p)s_t$ are $G$-steerable kernels, or matrix-valued functions: $K : \mathcal{S} \times \mathcal{A} \to \mathbb{R}^{d_\mathcal{A} \times d_\mathcal{S}}$ and $P : \mathcal{S} \times \mathcal{A} \to \mathbb{R}^{d_\mathcal{S} \times d_\mathcal{S}}$.*

**What tasks are suitable for Euclidean equivariance?** We find tasks that have dominated *global* Euclidean symmetry (change of reference frame) and less local symmetry can have relatively better parameter reduction, shown in Table 1. Tasks with more kinematic constraints make them harder to exploit Euclidean equivariance. For examples, for kinematic chains, it inherently has local coordinates, which makes it harder to use Euclidean symmetry w.r.t. the global reference frame. To better exploit symmetry there, it needs to explicitly consider constraints, such as the case in position-based (Tsai, 2017) or particle-based dynamics (Han et al., 2022).

---

[5]For simplicity, the theory omits an encoder from the state to latent space enc : $\mathcal{S} \to \mathcal{Z}$ in TD-MPC and MuZero style algorithms. In implementation, we follow TD-MPC that uses a learned encoder, which, when paired with an equivariant downstream network, helps learn symmetric representations. (Park et al., 2022).

**Interpretation and Examples.** The theory means that the linearized dynamics must satisfy a more restrictive set of conditions to be G-equivariant compared to the full, non-linear dynamics. It gives us a theoretical estimation of free parameters for *each* task, as well as improvement of sample efficiency when using (Jedra & Proutiere, 2021). We discuss further examples and computation of reduced dimensions in Appendix D. A toy example that illustrates how symmetry reduces the number of free parameters is moving a 3D particle (3D `PointMass`, Example 4 in Table 1): the matrix $A(p)$ (for $p \in \mathcal{S} \times \mathcal{A}$) will have dimension $6 \times 6$ but can be decomposed to $2 \times 2$ blocks of $3 \times 3$ sub-matrices, each only with 3 free parameters. Thus, for each orbit $p$, $A(p)$ and $B(p)$ only have $4 \times 3 = 12$ free parameters. Additionally, on a given orbit, $A(p)$ and $B(p)$ have explicit forms as shown in Figure 3. In summary, $G$-equivariance constrains the number of parameters in the dynamics and policy functions, enabling more sample-efficient learning.

## 4 SYMMETRY IN SAMPLING-BASED MODEL-BASED RL ALGORITHMS

In this section, after confirming the effectiveness of symmetry in planning, we develop an equivariant model-based RL algorithm for continuous action spaces to exploit continuous symmetry. To plan in continuous spaces, we require *sampling-based* methods such as MPPI (Williams et al., 2015; 2017b), extending them to preserve equivariance. We build on prior work (Zhao et al., 2022b) that used value-based planning on a discrete state space $\mathbb{Z}^2$ and discrete group $D_4$, extending this work to the continuous case. The idea is to ensure that the algorithm $a_t = \texttt{plan}(s_t)$ produces the same actions up to transformations, i.e., it is $G$-equivariant: $g \cdot a_t \equiv g \cdot \texttt{plan}(s_t) = \texttt{plan}(g \cdot s_t)$, as shown in Figure 1. The principle is applicable for MDPs with other symmetry groups.

### 4.1 COMPONENTS

We use TD-MPC (Hansen et al., 2022) as the backbone of our implementation and introduce their procedure and demonstrate how to incorporate symmetry into sampling-based planning algorithms.

- *Planning with learned models.* We use the MPPI (Model Predictive Path Integral) control method (Williams et al., 2015; 2016; 2017a;b), as adopted in TD-MPC (Hansen et al., 2022). We sample $N$ trajectories with horizon $H$ using the learned dynamics model, with actions from a learned policy, and estimate the expectation of total return.

- *Training models.* The learnable components in equivariant TD-MPC include: an `encoder` that processes input observation, `dynamics` and `reward` networks that simulate the MDP, and `value` and `policy` networks that guide the planning process.

- *Loss.* The only requirement is that loss is $G$-invariant. The loss terms in TD-MPC include value-prediction MSE loss and dynamics/reward-consistency MSE loss, which all satisfy invariance.

### 4.2 INTEGRATING SYMMETRY

Zhao et al. (2022b) consider how the Bellman operator transforms under symmetry transformation. For sampling-based methods, one needs to consider how the sampling procedure changes under symmetry transformation. Specifically, under a symmetry transformation, differently sampled trajectories must transform equivariantly. This is shown in Figure 1. The equivariance of the transition model in sampling-based approches to machine learning has also been studied in (Park et al., 2022). There are several components that need $G$-equivariance, and we discuss them step-by-step and illustrate them in Figure 1.

1. **dynamics and reward model.** In the definition of symmetry in Geometric MDPs (and symmetric MDPs (Ravindran & Barto, 2004; van der Pol et al., 2020b; Zhao et al., 2022b)) in Equation 1, the transition and reward functions are $G$-equivariant and $G$-invariant respectively. Therefore, in implementation, the transition network is deterministic and uses a $G$-equivariant MLP, and the reward network is constrained to be $G$-invariant. Additionally, in implementation, planning is typically performed in latent space, using a latent dynamics model $\bar{f}(z, a) = z'$. To do this, we require a $G$-equivariant encoder $h : \mathcal{S} \to \mathcal{Z}$, satisfying $\rho_{\mathcal{Z}}(g) \cdot h(s) = h(\rho_{\mathcal{S}}(g) \cdot s)$. We omit the encoder in our description below for notational simplicity.

2. **`value and policy` model.** The optimal value function produces a scalar for each state and is $G$-invariant, while the optimal policy function is $G$-equivariant (Ravindran & Barto, 2004). If we use $G$-equivariant transition and $G$-invariant reward networks in updating our value function $\mathcal{T}[V_\theta] = \sum_{\boldsymbol{a}} R_\theta(\boldsymbol{s}, \boldsymbol{a}) + \gamma \sum_{\boldsymbol{s}'} P_\theta(\boldsymbol{s}'|\boldsymbol{s}, \boldsymbol{a})V_\theta(\boldsymbol{s}')$, the learned value network $V_\theta$ will also satisfy the symmetry constraint. Similarly, we can extract an optimal policy from the value network, which is also $G$-equivariant (van der Pol et al., 2020b; Wang et al., 2021; Zhao et al., 2022b).

3. **`MPC` procedure.** We consider equivariance in the MPC procedure in two parts: `sample` trajectories from the MDP using learned models, and compute their `returns`, $\text{return}(\text{sample}(s, \theta))$. We discuss the invariance and equivariance of it in the next subsection.

We list the equivariance or invariance conditions that each network needs to satisfy. Alternatively, for scalar functions, we can also say they transform under *trivial* representation $\rho_0$ and are thus invariant. All modules are implemented via $G$-steerable equivariant MLPs: $\rho_{\text{out}}(g) \cdot y = \rho_{\text{out}}(g) \cdot \text{MLP}(x) = \text{MLP}(\rho_{\text{in}}(g) \cdot x)$.

$$\begin{aligned} f_\theta : \mathcal{S} \times \mathcal{A} \to \mathcal{S} : && \rho_\mathcal{S}(g) \cdot f_\theta(\boldsymbol{s}_t, \boldsymbol{a}_t) = f_\theta(\rho_\mathcal{S}(g) \cdot \boldsymbol{s}_t, \rho_\mathcal{A}(g) \cdot \boldsymbol{a}_t) && (7) \\ R_\theta : \mathcal{S} \times \mathcal{A} \to \mathbb{R} : && R_\theta(\boldsymbol{s}_t, \boldsymbol{a}_t) = R_\theta(\rho_\mathcal{S}(g) \cdot \boldsymbol{s}_t, \rho_\mathcal{A}(g) \cdot \boldsymbol{a}_t) && (8) \\ Q_\theta : \mathcal{S} \times \mathcal{A} \to \mathbb{R} : && Q_\theta(\boldsymbol{s}_t, \boldsymbol{a}_t) = Q_\theta(\rho_\mathcal{S}(g) \cdot \boldsymbol{s}_t, \rho_\mathcal{A}(g) \cdot \boldsymbol{a}_t) && (9) \\ \pi_\theta : \mathcal{S} \to \mathcal{A} : && \rho_\mathcal{A}(g) \cdot \pi_\theta(\cdot \mid \boldsymbol{s}_t) = \pi_\theta(\cdot \mid \rho_\mathcal{S}(g) \cdot \boldsymbol{s}_t) && (10) \end{aligned}$$

### 4.3 EQUIVARIANCE OF `MPC`

We analyze how to constrain the underlying MPC planner to be equivariant. We use MPPI (Model Predictive Path Integral) (Williams et al., 2015; 2017a), which has been used in TD-MPC for action selection. An MPPI procedure samples multiple $H$-horizon trajectories $\{\tau_i\}$ from the current state $\boldsymbol{s}_t$ using the learned models. We use `sample` to refer to the procedure: $\tau_i \equiv \text{sample}(\boldsymbol{s}_t; f_\theta, R_\theta, Q_\theta, \pi_\theta) = (\boldsymbol{s}_t, \boldsymbol{a}_t, \boldsymbol{s}_{t+1}, \boldsymbol{a}_{t+1}, \ldots, \boldsymbol{s}_{t+H})$. Another procedure `return` computes the accumulated return, evaluating the value of a trajectory for top-$k$ trajectories:

$$\text{return}(\tau) = \mathbb{E}_\tau \left[ \gamma^H Q_\theta(\boldsymbol{s}_H, \boldsymbol{a}_H) + \sum_{t=0}^{H-1} \gamma^t R_\theta(\boldsymbol{s}_t, \boldsymbol{a}_t) \right] = \mathbb{E}_\tau \left[ U(\boldsymbol{s}_{1:H}, \boldsymbol{a}_{1:H-1}) \right] \quad (11)$$

A trajectory is transformed element-wise by a transformation $g$: $g \cdot \tau_i = (g \cdot \boldsymbol{s}_t, g \cdot \boldsymbol{a}_t, g \cdot \boldsymbol{s}_{t+1}, g \cdot \boldsymbol{a}_{t+1}, \ldots, g \cdot \boldsymbol{s}_{t+H})$. However, since $\mu$ and $\sigma$ in action sampling are *not state-dependent*, the MPPI `sample` does not exactly preserve equivariance: rotating the input does not *deterministically* guarantee a rotated output.

We propose a strategy to fix it. We consider the simplified case with a single time step, so the sampling draws $N$ actions from a random Gaussian distribution $\mathcal{N}(\mu, \sigma^2 I)$, denoted as $\mathbb{A} = \{\boldsymbol{a}_i\}_{i=1}^N$. The return is simply $Q(s, a)$. Assuming we only select the best trajectory ($K = 1$), we require the following procedure to be equivariant: $a_0 = \arg\max_a Q(s_0, a)$. In other words, if we rotate state $g \cdot s_0$, the selected action is also rotated $g \cdot a_0$. Thus, a simple strategy is to augment the action sampling with $G$: $G\mathbb{A} = \{g \cdot a_i \mid g \in G\}_{i=1}^N$. We indicate that sampling strategy as $G\text{-sample}$.

**Proposition 5** *The `return` procedure is $G$-invariant, and the $G$-augmented $G\text{-sample}$ procedure that augment $\mathbb{A}$ using transformation in $G$ is $G$-equivariant when $K = 1$.*

We further explain in Appendix E. In summary, for sampling and computing return, they satisfy the following conditions, indicating that the procedure $\text{return}(G\text{-sample}(s, \theta))$ is invariant, i.e., not changed under group transformation for any $g$. We use $\text{return}(\tau_i)$ to indicate the return of a specific trajectory $\tau_i$ and $g \cdot \tau_i$ to denote group action on it.

$$\begin{aligned} G\text{-sample} : \boldsymbol{s}_t, \theta \mapsto \tau_i : && g \cdot \tau_i \sim G\text{-sample}(g \cdot \boldsymbol{s}_t; f_\theta, R_\theta, Q_\theta, \pi_\theta) && (12) \\ \text{return} : \tau_i \mapsto \mathbb{R} : && \text{return}(\tau_i) = \text{return}(g \cdot \tau_i) && (13) \end{aligned}$$

## 5 EVALUATION: SAMPLING-BASED PLANNING

In this section, we present the setup and results for our proposed sampling-based planning algorithm: equivariant version of TD-MPC. The additional details and results are available in Appendix F.

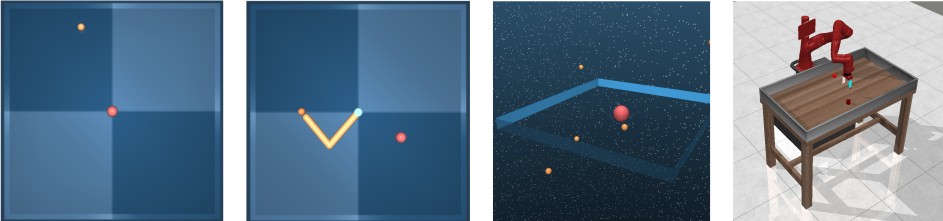

Figure 4: Tasks used in experiments: (1) `PointMass` in 2D, (2) `Reacher`, (3) Customized 3D version of `PointMass` with multiple particles to control, and (4) MetaWorld task to reach an object with gripper.

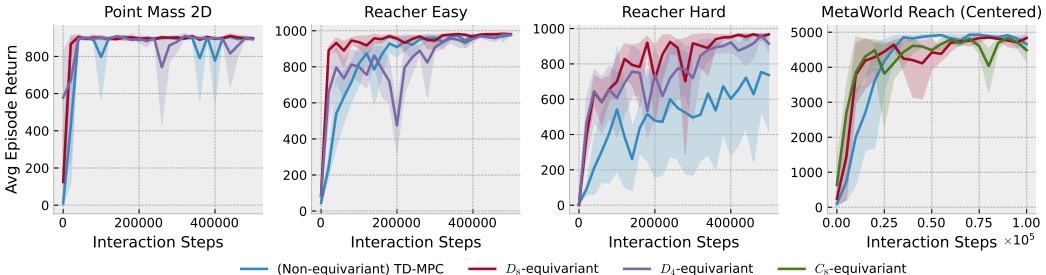

Figure 5: Results on 2D `PointMass`, `Reacher`, and MetaWorld `Reach` task.

**Tasks.** We verify the algorithm on a few selected and customized tasks using DeepMind Control suite (DMC) (Tassa et al., 2018), visualized in Figure 4. One task is 2D particle moving in $\mathbb{R}^2$, `PointMass`. We customize tasks based on it: (1) 3D particle moving in $\mathbb{R}^3$ (disabled gravity), and (2) 3D $N$-point moving that has several particles to control simutaneously. The goal is to move particle(s) to a target position. We also experiment with tasks on a two-link arm, `Reacher` (easy and hard), where the goal is to move the end-effector to a random position in a plane. `Reacher Easy` and `Hard` are top-down where the goal is to reach a random 2D position. If we rotate the MDP, the angle between the first and second links is not affected, i.e., it is $G$-invariant. The first joint and the target position are transformed under rotation, so we set it to $\rho_1$ standard representation (2D rotation matrices). The complete state and action representations are given in Table 1. The system has O(2) rotation and also reflection symmetry, hence we use $D_8$ and $D_4$ groups.

We also use MetaWorld tabletop manipulation (Yu et al., 2019). The action space is 3D gripper movement $(\Delta x, \Delta y, \Delta z)$ and 1D openness. The state space (1) gripper position, (2) 3D position plus 4D quaternion of at most 2 relevant objects, (3) 3D randomized goal position, depending on tasks. If we consider tasks with gravity, the MDP itself should exhibit SO(2) symmetry about the gravity axis. We make the origin at workspace center and the gripper initialized at the origin, so the task respects rotation equivariance around the origin.

**Experimental setup.** We compare against the non-equivariant version of TD-MPC (Hansen et al., 2022). Here, we by default make all components equivariant as described in the algorithm section. In Sec F.3, we include ablation studies for disabling or enabling each equivariant component. The training procedure follows TD-MPC (Hansen et al., 2022). We use the state as input and for equivariant TD-MPC, we divide the orignal hidden dimension by $\sqrt{N}$, where $N$ is the group order, to keep the number of parameters roughly equal between the equivariant and non-equivariant versions. We mostly follow the original hyperparameters except for `seed_steps`. We use 5 random seeds for each method.

**Algorithm setup: equivariance.** We use discretized subgroups in implementing $G$-equivariant MLPs with the `escnn` package (Weiler & Cesa, 2021), which are more stable and easier to implement than continuous equivariance. For the 2D case, we use O(2) subgroups: dihedral groups $D_4$ and $D_8$ (4 or 8 rotation components), or rotation group $C_8$ (45° rotations). For the 3D case, we use the Icosahedral and Octahedral groups, which are finite subgroups of SO(3) with orders 60 and 24 respectively.

**Results.** In Figures 5 and 6, we show the reward curves and demonstrate that our equivariant methods can reach near-optimal performance 2x or 3x faster in terms of training interaction steps for

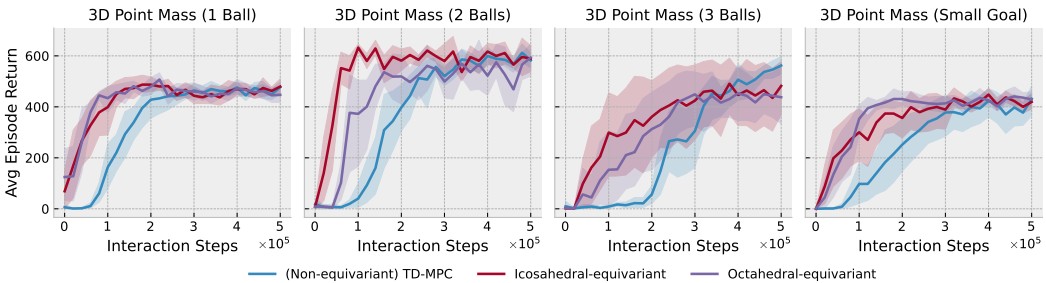

Figure 6: Results on a set of customized 3D $N$-ball `PointMass` tasks, with $N = 1, 2, 3$, and customized a 3D `PointMass` with smaller target.

several tasks. Recall the examples of Geometric MDPs, our theory not only motivates the algorithm design, but also gives an estimation of what tasks can benefit from (continuous) symmetry. The results justify the theoretical estimation on improvement of sample efficiency: less free parameters result in better regret bound and faster learning, as shown for LQR (Jedra & Proutiere, 2021).

In terms of the actual results, the default `PointMass` 2D version seems easy to solve, while the $D_8$-equivariant version learns slightly faster. For `Reacher`, as shown in Figure 5, $D_8$ outperforms the non-equivariant TD-MPC by noticeable margins, especially on the `Hard` domain. $D_4$ is slightly worse than $D_8$ but still better than the baseline. The rightmost subfigure shows a MetaWorld task `Reach`, which is to reach a button on a desk using the parallel gripper. We add $\mathrm{SO}(2)$ equivariance to the algorithm about the gravity axis and evaluate the $C_8$ and $D_8$-equivariant versions, which both give more efficient learning.

On `Reacher` tasks, we also compare against a *planning-free* baseline by removing MPPI planning with the learned model and only keep policy learning, shown in Fig 11, which is effectively similar to the DDPG algorithm (Lillicrap et al., 2016).

We design a set of harder 3D versions of `PointMass` and use $\mathrm{SO}(3)$ subgroups to implement 3D equivariant versions of TD-MPC, using Icosahedral- and Octahedral-equivariant MLPs. Figure 6 shows $N = 1, 2, 3$ balls in 3D `PointMass`, and the rightmost figure shows 1-ball 3D version with smaller target (0.02 compared to 0.03 in $N$-ball version). We find the Icosahedral (order 60) equivariant TD-MPC always learns faster and uses fewer samples to achieve best rewards, compared to the non-equivariant version. The Octahedral (order 24) equivariant version also performs similarly. The best absolute rewards in the 1-ball case is interestingly lower than 2- and 3-ball casees, which may be caused by higher possible return due to the presence of 2 or 3 balls that can reach the goal.

With higher-order 2D discrete subgroups, the performance plateaus but computational costs increase, so we use up to $D_8$. We also find TD-MPC is especially sensitive to a hyperparameter `seed_steps` that controls the number of warmup trajectories. In contrast, our equivariant version is robust to it and sometimes learn better with less warmup. In the shown curves, we do not use warmup across non-equivariant and equivariant ones and present additional results in Appendix F.

## 6 CONCLUSION AND DISCUSSION

In conclusion, we underscore the value of Euclidean symmetry in model-based RL algorithms. We define a subclass of MDPs, Geometric MDPs, prevalent in robotics, which exhibit additional structure. The linearized approximation of these MDPs adheres to steerable kernel constraints, substantially reducing parameter space. Drawing from this, we developed a model-based RL algorithm, utilizing Euclidean symmetry, that outperforms standard techniques in common RL benchmarks. This is the first method considering the importance of equivariance in sampling-based RL methods. It contributes to a deeper understanding of symmetry in RL algorithms and offers insights for future research. However, our theory and experience also show that while Euclidean symmetry can bring significant savings in parameters, it does not always offer practical benefits for some tasks with local coordinates. For instance, locomotion tasks do not greatly benefit from it. Also, our approach assumes the symmetry group is known, typically determined by the robot workspace dimension, usually 3D.

