CONTENTS

## A  OUTLINE

The appendix is organized as follows: (1) additional discussion, including related work and theoretical background, (2) theory, derivation, and proofs, (3) implementation details and further empirical results, and (4) additional mathematical background.

## B  ADDITIONAL DISCUSSION

### B.1  DISCUSSION: SYMMETRY IN DECISION-MAKING

In this work, we study the Euclidean symmetry $\mathrm{E}(d)$ from geometric transformations between *reference frames*. This is a specific set of symmetries that an MDP can have – isometric transformations

of Euclidean space $\mathbb{R}^d$, such as the distance is preserved. This can be viewed as a special case under the framework of MDP homomorphism, where symmetries relate two different MDPs via MDP *homomorphism* (or more strictly, *isomorphism*). We refer the readers to (Ravindran & Barto, 2004) for more details. We also discuss symmetry in other related fields.

Classic planning algorithms and model checking have leveraged the use of symmetry properties, (Fox & Long, 1999; 2002; Pochter et al., 2011; Domshlak et al.; Shleyfman et al., 2015; Sievers et al., 2015; Sievers; Sievers et al., 2019; Fiser et al., 2019) as evident from previous research. In particular, Zinkevich & Balch (2001) demonstrate that the value function of an MDP is invariant when symmetry is present. However, the utilization of symmetries in these algorithms presents a fundamental problem since they involve constructing equivalence classes for symmetric states, which is difficult to maintain and incompatible with differentiable pipelines for representation learning. Narayanamurthy & Ravindran (2008) prove that maintaining symmetries in trajectory rollout and forward search is intractable (NP-hard). To address the issue, recent research has focused on state abstraction methods such as the coarsest state abstraction that aggregates symmetric states into equivalence classes studied in MDP homomorphisms and bisimulation (Ravindran & Barto, 2004; Ferns et al., 2004; Li et al., 2006). However, the challenge lies in that these methods typically require perfect MDP knowledge and do not scale well due to the complexity of constructing and maintaining abstraction mappings (van der Pol et al., 2020a). To deal with the difficulties of symmetry in forward search, recent studies have integrated symmetry into reinforcement learning based on MDP homomorphisms (Ravindran & Barto, 2004), including van der Pol et al. (2020a) that integrate symmetry through an equivariant policy network. Furthermore, Mondal et al. (2020) previously applied a similar idea without using MDP homomorphisms. Park et al. (2022) learn equivariant transition models, but do not consider planning, and Zhao et al. (2022a) focuses on permutation symmetry in object-oriented transition models. Recent research by (Zhao et al., 2022b; 2023b) on 2D discrete symmetry on 2D grids has used a value-based planning approach.

There are some benefits of explicitly considering symmetry in continuous control. The possibility of hitting orbits is negligible, so there is no need for orbit-search on symmetric states in forward search in continuous control. Additionally, the planning algorithm implicitly plans in a smaller continuous MDP $\mathcal{M}/G$ (Ravindran & Barto, 2004). Furthermore, from equivariant network literature (Elesedy & Zaidi, 2021), the generalization gap for learned equivariant policy and value networks are smaller, which allows them to generalize better.

## B.2 ADDITIONAL RELATED WORK

**Geometric deep learning and equivariant networks.** Geometric deep learning is a field that examines how to maintain geometric properties, such as symmetry and curvature, in data analysis (Bronstein et al., 2021). To preserve symmetries in data, researchers have developed equivariant neural networks. For instance, Cohen & Welling (2016c) introduced G-CNNs, followed by Steerable CNNs (Cohen & Welling, 2016d), which generalize scalar feature fields to vector fields and the induced representations. Moreover, Kondor & Trivedi (2018); Cohen et al. (2020b) have studied the theory on equivariant maps and convolutions for scalar fields through trivial representations and vector fields through induced representations, respectively. Furthermore, Weiler & Cesa (2021) propose $E(2)$-CNN, a method for solving kernel constraints for $E(2)$ and its subgroups, by decomposing into irreducible representations. Researchers have also explored how to use steerable features to maintain symmetries in deep learning models. For example, Brandstetter et al. (2022) developed steerable message-passing GNNs that use equivariant steerable features, while Satorras et al. (2021) use only invariant scalar features to build $E(n)$-equivariant graph networks. The idea of steerable features is further developed in Brandstetter et al. (2022), who propose steerable message passing graph networks for 3D space.

**Learned dynamics model for interactive system.** A framework for learning interaction dynamics between objects in a scene was proposed by Battaglia et al. (2016). This approach is based on relational inductive biases that consider the relationships among objects. Battaglia et al. (2018) later expanded on this framework by introducing relational networks that learn the dynamics between objects within a graph-based representation. Similarly, Sanchez-Gonzalez et al. (2018) developed a graph neural network model for physics simulation to learn dynamics in a graph-based representation. Furthermore, Li et al. (2019) introduced a particle-based dynamics network that focuses on

Table 2: (Copied from main text) Examples of geometric MDPs. $G$ denotes the MDP symmetry group. $\mathcal{S}$ denotes the MDP state space. $\mathcal{A}$ denotes the MDP action space. We can quantitatively measure the saving of equivariance. "Images" refers to panoramic egocentric images $\mathbb{Z}^2 \to \mathbb{R}^{H \times W \times 3}$. $\circ$ denotes group element composition. We list the quotient space $\mathcal{S}/G$ to give intuition on saving. $Gx = \{g \cdot x \mid g \in G\}$ column shows the $G$-orbit space of $\mathcal{S}$ ($\cong$ denotes isomorphic to).

| ID | $G$ | $\mathcal{S}$ | $\mathcal{A}$ | $\mathcal{S}/G$ | $Gx$ | Task |
|---|---|---|---|---|---|---|
| 1 | $C_4$ | $\mathbb{Z}^2$ | $C_4$ | $\mathbb{Z}^2/C_4$ | $C_4$ | 2D Path Planning (Tamar et al., 2016) |
| 2 | $C_4$ | Images | $C_4$ | $\mathbb{Z}^2/C_4$ | $C_4$ | 2D Visual Navigation (Zhao et al., 2022b) |
| 3 | SO(2) | $\mathbb{R}^2$ | $\mathbb{R}^2$ | $\mathbb{R}^+$ | $S^1$ | 2D Continuous Navigation |
| 4 | SO(3) | $\mathbb{R}^3 \times \mathbb{R}^3$ | $\mathbb{R}^3$ | $\mathbb{R}^+ \times \mathbb{R}^3$ | $S^2$ | 3D Free particle (with velocity) |
| 5 | SO(3) | $\mathbb{R}^3 \rtimes$ SO(3) | $\mathbb{R}^3 \times \mathbb{R}^3$ | $\mathbb{R}^+ \times \mathbb{R}^3$ | $S^2$ | Moving 3D Rigid Body |
| 6 | SO(2) | SO(2) | $\mathbb{R}^2$ | $\{e\}$ | $S^1$ | Free Particle on SO(2) $\cong S^1$ manifold |
| 7 | SO(3) | SO(3) | $\mathbb{R}^3$ | $\{e\}$ | $S^2$ | Free Particle on SO(3) (Teng et al., 2023) |
| 8 | SO(2) | SE(2) | SE(2) | $\mathbb{R}^2$ | $S^1$ | Top-down grasping (Zhu et al., 2022) |
| 9 | SO(2) | $(S^1)^2 \times (\mathbb{R}^2)^2$ | $\mathbb{R}^2$ | $S^1 \times (\mathbb{R}^2)^2$ | $S^1$ | Two-arm manipulation (Tassa et al., 2018) |

the physical interactions between particles in a simulation. This approach enables the generation of realistic animations and predictions of future states.

## B.3 LIMITATIONS AND FUTURE WORK

Although Euclidean symmetry group is infinite and seems huge, it does not guarantee significant performance gain in all cases. Our theory helps us understand when such Euclidean symmetry may not be very beneficial The key issue is that when a robot has kinematic constraints, Euclidean symmetry does not change those features, which means that equivariant constraints cannot share parameters and reduce dimensions. We empirically show this on using local vs. global reference frame in the additional experiment in Sec F. For further work, one possibility is to explicit consider constraints while keep using global positions.

## B.4 ILLUSTRATION AND EXAMPLES OF GEOMETRIC MDPS

In Figure 2, we present visual examples of Geometric MDPs and non-geometric MDPs in both discrete and continuous cases. Geometric MDP examples include moving a point robot in a 2D continuous space ($\mathbb{R}^2$, Example 3 in Table 2) or a discrete space ($\mathbb{Z}^2$, Example 1 (Tamar et al., 2016)), which is the abstraction of 2D discrete or continuous navigation. Table 2 includes more relevant examples. We use visual navigation over a 2D grid ($\mathbb{Z}^2 \rtimes C_4$, Example 2 (Lee et al., 2018; Zhao et al., 2022b)) as another example of a Geometric MDP. In this example, each position in $\mathbb{Z}^2$ and orientation in $C_4$ has an image in $\mathbb{R}^{H \times W \times 3}$, which is a *feature map* $\mathbb{Z}^2 \rtimes C_4 \to \mathbb{R}^{H \times W \times 3}$. The agent only navigates on the 2D grid $\mathbb{Z}^2$ (potentially with an orientation of $C_4$), but not the raw pixel space. Example (4) extends to the continuous 3D space and also include linear velocity $\mathbb{R}^3$. Alternatively, we can consider (5) moving a rigid body with SO(3) rotation. In (6) and (7), we consider moving free particle positions on SO(2), SO(3), which are examples of optimal control *on manifold* in (Lu et al., 2023; Teng et al., 2023). Here, $G = $ SO(3) acts on $\mathcal{S} = $ SO(3) by group composition. (8) top-down grasping needs to predict SE(2) action on grasping an object on plane with SE(2) pose. It additionally has translation symmetry, so the state space is technically SE(2)/SE(2) = $\{e\}$. (9) is the `Reacher` task that we studies later, which controls a two-joint arm. It is easy to see that because two links are connected, kinematic constraints come in and equivariance does not save so much. Additionally, Example (3) is later implemented as `PointMass`, which is (8) top-down grasping without SO(2) rotation.

## B.5 CONTINUOUS GROUP ACTIONS FOR GEOMETRIC MDPS

If the $G$-actions on $\mathcal{S}$ and $\mathcal{A}$ are continuous, there is an interesting geometric interpretation based on fiber bundle theory (Husemöller, 2013). This requirement is not mandatory for implementation but allows for more rigorous theoretical results. Continuous symmetries correspond to conservation laws, while discrete (non-differentiable) symmetries do not have corresponding conservation laws

(Zee, 2016). The linearized dynamics of a system are much more constrained (i.e., have fewer free parameters) when the system has continuous $G$-actions. If the group $G$ is a *compact* Lie group and has continuous $G$-actions, there exists a map $p : \mathcal{S} \times \mathcal{A} \mapsto \mathcal{B}$ that projects the state-action space $\mathcal{S} \times \mathcal{A}$ to a lower dimensional base space $\mathcal{B}$ (Cohen et al., 2020b). The existence and smoothness of the projection $p$ can be established using principal bundle theory. See Appendix D for more detail.

## C  MATHEMATICAL BACKGROUND

### C.1  BACKGROUND FOR REPRESENTATION THEORY AND $G$-STEERABLE KERNELS

We establish some notation and review some elements of group theory and representation theory. For a comprehensive review of group theory and representation theory, please see (Serre, 2005). The identity element of any group $G$ will be denoted as $e$. We will always work over the field $\mathbb{R}$ unless otherwise specified.

### C.2  GROUP DEFINITION

A group is a non-empty set equipped with an associative binary operation $\cdot : G \times G \to G$ where $\cdot$ satisfies

$$\text{Existence of identity: } \exists e \in G, \text{ s.t. } \forall g \in G, \ e \cdot g = g \cdot e = g$$

$$\text{Existence of inverse: } \forall g \in G, \exists g^{-1} \in G \text{ s.t. } g \cdot g^{-1} = g^{-1} \cdot g = e$$

For a complete reference on group theory, please see Zee (2016).

#### C.2.1  GROUP REPRESENTATIONS

A group is an abstract object. Oftentimes, when working with groups, we are most interested in group *representations*. Let $V$ be a vector space over $\mathbb{C}$. A *representation* $(\rho, V)$ of $G$ is a map $\rho : G \to \text{Hom}[V, V]$ such that

$$\forall g, g' \in G, \ \forall v \in V, \quad \rho(g \cdot g')v = \rho(g) \cdot \rho(g')v$$

Concisely, a group representation is a embedding of a group into a set of matrices. The matrix embedding must obey the multiplication rule of the group. Over $\mathbb{R}$ and $\mathbb{C}$ all representations break down into irreducible representations Serre (2005). We will denote the set of irreducible representations of a group $G$ and $\hat{G}$.

#### C.2.2  GROUP ACTIONS

Let $\Omega$ be a set. A group action $\Phi$ of $G$ on $\Omega$ is a map $\Phi : G \times \Omega \to \Omega$ which satisfies

$$\text{Identity: } \forall \omega \in \Omega, \quad \Phi(e, \omega) = \omega$$

$$\text{Compositionality: } \forall g_1, g_2 \in G, \ \forall \omega \in \Omega, \quad \Phi(g_1 g_2, \omega) = \Phi(g_1, \Phi(g_2, \omega))$$

We will often suppress the $\Phi$ function and write $\Phi(g, \omega) = g \cdot \omega$.

$$
\begin{array}{ccc}
\Omega & \xrightarrow{\Psi} & \Omega' \\
\downarrow{\scriptstyle \Phi(g,\cdot)} & & \downarrow{\scriptstyle \Phi'(g,\cdot)} \\
\Omega & \xrightarrow{\Psi} & \Omega'
\end{array}
$$

Figure 7: Commutative Diagram For $G$-equivariant function: Let $\Phi(g, \cdot) : G \times \Omega \to \Omega$ denote the action of $G$ on $\Omega$. Let $\Phi'(g, \cdot) : G \times \Omega' \to \Omega'$ denote the action of $G$ on $\Omega'$ The map $\Psi : \Omega \to \Omega'$ is $G$-equivariant if and only if the following diagram is commutative for all $g \in G$.

Let $G$ have group action $\Phi$ on $\Omega$ and group action $\Phi'$ on $\Omega'$. A mapping $\Psi : \Omega \to \Omega'$ is said to be $G$-equivariant if and only if

$$\forall g \in G, \forall \omega \in \Omega, \quad \Psi(\Phi(g, \omega)) = \Phi'(g, \Psi(\omega)) \tag{14}$$

Diagrammatically, $\Psi$ is $G$-equivariant if and only if the diagram C.2.2 is commutative.

$G$-**Intertwiners**  Let $(\rho, V)$ and $(\sigma, W)$ be two $G$-representations. The set of all $G$-equivariant linear maps between $(\rho, V)$ and $(\sigma, W)$ will be denoted as

$$\mathrm{Hom}_G[(\rho, V), (\sigma, W)] = \{\Phi \mid \Phi : V \to W, \ \forall g \in G, \ \Phi(\rho(g)v) = \sigma(g)\Phi(v)\}$$

$\mathrm{Hom}_G$ is a vector space over $\mathbb{C}$. When the linear maps are restricted to be real, $\mathrm{Hom}_G$ forms a vector space over $\mathbb{R}$. A linear map $\Phi \in \mathrm{Hom}_G[(\rho, V), (\sigma, W)]$ is said to *intertwine* the representations $(\rho, V)$ and $(\sigma, W)$. An intertwiner $\Phi$ is a map that makes the diagram C.2.2 commutative.

$$
\begin{array}{ccc}
(\rho, V) & \xrightarrow{\ \Phi\ } & (\sigma, W) \\
\downarrow{\scriptstyle \rho(g)} & & \downarrow{\scriptstyle \sigma(g)} \\
(\rho, V) & \xrightarrow{\ \Phi\ } & (\sigma, W)
\end{array}
$$

Figure 8: Commutative Diagram For $G$-intertwiner. The map $\Psi :\in \mathrm{Hom}_G[(\rho, V), (\sigma, W)]$ if and only if the following diagram is commutative for all $g \in G$.

Computing a basis for the vector space $\mathrm{Hom}_G[(\rho, V), (\sigma, W)]$ is an important procedure in the theory of steerable kernels (Cohen & Welling, 2016a).

### C.2.3   CLEBSCH-GORDON COEFFICIENTS

Let $G$ be a compact group. Let $\hat{G}$ be the irreducible representations of $G$ Let $(\rho, V)$ and $(\sigma, W)$ be irreducible representations of $G$. The tensor product representation will not in general be irreducible and

$$(\rho, V) \otimes (\sigma, W) = \bigoplus_{\tau \in \hat{G}} c_{\sigma,\rho}^{\tau} (\tau, V_\tau)$$

where $c_{\sigma,\rho}^{\tau}$ are the Clebsch-Gordon multiplicities which count the number of copies of the irreducible $(\tau, V_\tau)$ in the tensor product representation $(\rho, V) \otimes (\sigma, W)$. Clebsch-Gordon Coefficients $C_{\rho_1 \rho_2}^{\tau}$ are the coefficients of the representation $(\tau, V_\tau)$ in the tensor product basis. Specifically, let

$$|\tau i_\tau\rangle = \sum_{j_1=1}^{d_1} \sum_{j_2=1}^{d_2} \underbrace{\langle \rho_1 j_1, \rho_2 j_2 | \tau i_\tau \rangle}_{(C_{\rho_1 \rho_2}^{\tau})_{i_\tau, j_1 j_2}} |\rho_1 j_1, \rho_2 j_2\rangle$$

Clebsch-Gordon coefficients are an integral part of the general solution to steerable kernel constraint Lang & Weiler (2020a).

### C.2.4   CHARACTERIZATION OF STEERABLE KERNELS ON HOMOGENEOUS SPACES

We briefly summarize the results of (Lang & Weiler, 2020a). Let $X$ be a homogeneous space of a compact group $G$. Let $(\sigma, V_\sigma) \in \hat{G}$ and $(\rho, V_\rho) \in \hat{G}$ be two $G$-irreducibles. Consider the kernel constraint

$$K(g \cdot x) = \sigma(g) K(x) \rho(g^{-1})$$

where $K : X \to \mathrm{Hom}[V_\rho, V_\sigma]$. Then, there exists a set of generalized spherical harmonics $Y_{\rho,k}^i :$ $X \to \mathbb{R}$ where $\rho \in \hat{G}$ is a $G$-irreducible and the index $i \in \{1, 2, ..., d_\rho\}$ and $k \in \{1, 2, ..., m_\rho\}$ where $m_\rho \leq d_\rho$ is called the muplicity which satisfy the relation

$$\forall g \in G, \quad Y_{\rho,k}^i(g^{-1} \cdot x) = \sum_{i'=1}^{d_\rho} \rho_{ii'}(g) Y_{\rho,k}^{i'}(x)$$

The set of $Y_{\rho,k}^i$ form a basis for all square integrable functions on $X$.

Let us define

$$K_{\sigma\rho}^{\tau ks}(x) = \sum_{i_\tau=1}^{d_\tau} \sum_{j_\sigma=1}^{d_\sigma} \sum_{i_\rho=1}^{d_\rho} |\sigma j_\sigma\rangle \underbrace{\langle s, \sigma j_\sigma | \tau i_\tau, \rho i_\rho \rangle}_{\text{Clebsch-Gordon}} \underbrace{Y_{\rho,k}^{i_\rho}(x)}_{\text{harmonics}} \langle \rho i_\rho |$$

Then, using the main result of (Lang & Weiler, 2020a), the matrices $K_{\sigma\rho}^{\tau ks}(x)$ form a basis for the space of $G$-steerable kernels with input representation $\rho$ and output representation $\sigma$. Then, the kernel $K$ can be written in the form

$$K_{\sigma\rho}(x) = \sum_{\tau \in \hat{G}} \sum_{k=1}^{m_\rho} \sum_{s=1}^{m_\sigma} c^{\tau ks} K_{\sigma\rho}^{\tau ks}(x)$$

where $c_{\tau ks} \in \mathrm{Hom}_G[(\sigma, V_\sigma), (\sigma, V_\sigma)]$ is a $(\sigma, V_\sigma)$-endomorphism. The total number of free parameters in $K_{\sigma\rho}$ is

$$\dim K_{\sigma\rho} = m_\rho m_\sigma \sum_{\tau \in \hat{G}} C_{\tau\rho}^\sigma \times \dim \mathrm{Hom}_G[(\sigma, V_\sigma), (\sigma, V_\sigma)] \leq 4 m_\rho m_\sigma \sum_{\tau \in \hat{G}} c_{\tau\rho}^\sigma$$

which depends on both multiplicity $m_\tau$ of the homogeneous space $X$ and the Clebsch-Gordon Coefficients $c_{\tau\rho}^\sigma$ of the group $G$.

## D  THEORY AND PROOFS

The section is organized as follows. We first give the proofs to Theorem 1 and 2. Then, we discuss how we linearize dynamics of a Geometric MDP and $G$-steerable kernels in detail. The goal of the theory is to show that, in linearized case, Euclidean symmetry can provably *reduce* number of free parameters and the dimensions of the solution space. Under RL setup with *unknown* dynamics (and cost) function, the Euclidean equivariance constraints then potentially bring significant benefit because of *less parameters*.

### D.1  THEOREM 1 AND 2: EQUIVARIANCE IN GEOMETRIC MDPS

***Theorem 1*** *The Bellman operator of a GMDP is equivariant under Euclidean group* $\mathrm{E}(d)$.

*Proof.* The Bellman (optimality) operator is defined as

$$\mathcal{T}[V](\boldsymbol{s}) := \max_{\boldsymbol{a}} R(\boldsymbol{s}, \boldsymbol{a}) + \int d\boldsymbol{s}' P(\boldsymbol{s}' \mid \boldsymbol{s}, \boldsymbol{a}) V(\boldsymbol{s}'), \tag{15}$$

where the input and output of the Bellman operator are both value function $V : \mathcal{S} \to \mathbb{R}$. The theorem directly generalizes to $Q$-value function.

Under group transformation $g$, a feature map (field) $f : X \to \mathbb{R}^{c_{\text{out}}}$ is transformed as:

$$[L_g f](x) = [f \circ g^{-1}](x) = \rho_{\text{out}}(g) \cdot f\left(g^{-1}x\right), \tag{16}$$

where $\rho_{\text{out}}$ is the $G$-representation associated with output $\mathbb{R}^{c_{\text{out}}}$. For the *scalar* value map, $\rho_{\text{out}}$ is identity, or trivial representation.

For any group element $g \in \mathrm{E}(d) = \mathbb{R}^d \rtimes \mathrm{O}(d)$, we transform the Bellman (optimality) operator step-by-step and show that it is equivariant under $\mathrm{E}(d)$:

$$L_g[\mathcal{T}[V]](\boldsymbol{s}) \overset{(1)}{=} \mathcal{T}[V](g^{-1}\boldsymbol{s}) \tag{17}$$

$$\overset{(2)}{=} \max_{\boldsymbol{a}} R(g^{-1}\boldsymbol{s}, \boldsymbol{a}) + \int d\boldsymbol{s}' \cdot P(\boldsymbol{s}' \mid g^{-1}\boldsymbol{s}, \boldsymbol{a}) V(\boldsymbol{s}') \tag{18}$$

$$\overset{(3)}{=} \max_{\bar{\boldsymbol{a}}} R(g^{-1}\boldsymbol{s}, g^{-1}\bar{\boldsymbol{a}}) + \int d(g^{-1}\bar{\boldsymbol{s}}) \cdot P(g^{-1}\bar{\boldsymbol{s}} \mid g^{-1}\boldsymbol{s}, g^{-1}\bar{\boldsymbol{a}}) V(g^{-1}\bar{\boldsymbol{s}}) \tag{19}$$

$$\overset{(4)}{=} \max_{\bar{\boldsymbol{a}}} R(\boldsymbol{s}, \bar{\boldsymbol{a}}) + \int d(g^{-1}\bar{\boldsymbol{s}}) \cdot P(\bar{\boldsymbol{s}} \mid \boldsymbol{s}, \boldsymbol{a}) V(g^{-1}\bar{\boldsymbol{s}}) \tag{20}$$

$$\overset{(5)}{=} \max_{\bar{\boldsymbol{a}}} R(\boldsymbol{s}, \bar{\boldsymbol{a}}) + \int d\bar{\boldsymbol{s}} \cdot P(\bar{\boldsymbol{s}} \mid \boldsymbol{s}, \boldsymbol{a}) V(g^{-1}\bar{\boldsymbol{s}}) \tag{21}$$

$$\overset{(6)}{=} \mathcal{T}[L_g[V]](\boldsymbol{s}) \tag{22}$$

For each step:

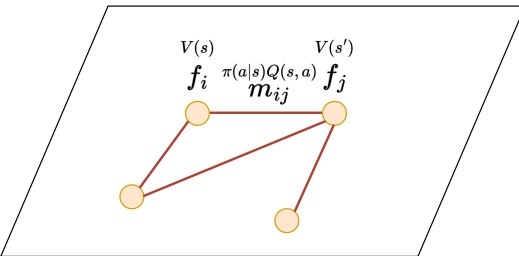

Figure 9: Demonstrate the idea of implementing value iteration with geometric message passing.

- (1) By definition of the (left) group action on the feature map $V : \mathcal{S} \to \mathbb{R}$, such that $g \cdot V(\boldsymbol{s}) = \rho_0(g)V(g^{-1}\boldsymbol{s}) = V(g^{-1}\boldsymbol{s})$. Because $V$ is a scalar feature map, the output transforms under trivial representation $\rho_0(g) = \mathrm{Id}$.

- (2) Substitute in the definition of Bellman operator.

- (3) Substitute $\boldsymbol{a} = g^{-1}(g\boldsymbol{a}) = g^{-1}\bar{\boldsymbol{a}}$. Also, substitute $g^{-1}\bar{\boldsymbol{s}} = \boldsymbol{s}'$.

- (4) Use the symmetry properties of Geometric MDP: $P(\boldsymbol{s}' \mid \boldsymbol{s}, \boldsymbol{a}) = P(g \cdot \boldsymbol{s} \mid g \cdot \boldsymbol{s}, g \cdot \boldsymbol{a})$ and $R(\boldsymbol{s}, \boldsymbol{a}) = R(g \cdot \boldsymbol{s}, g \cdot \boldsymbol{a})$.

- (5) Because $g \in \mathrm{E}(d)$ is isometric transformations (translations $\mathbb{R}^d$, rotations and reflections $\mathrm{O}(d)$) and the state space carries group action, the measure $ds$ is a $G$-invariant measure $d(gs) = ds$. Thus, $d\bar{\boldsymbol{s}} = d(g^{-1}\bar{\boldsymbol{s}})$.

- (6) By the definition of the group action on $V$.

The proof requires the MDP to be a Geometric MDP with Euclidean symmetry and the state space carries a group action of Euclidean group. Therefore, the Bellman operator of a Geometric MDP is $\mathrm{E}(d)$-equivariant. Additionally, we can also parameterize the dynamics and reward functions with neural networks, and the learned Bellman operator is also equivariant.

The proof is analogous to the case in (Zhao et al., 2022b), where the symmetry group is $p4m = \mathbb{Z}^2 \rtimes D_4$, which is a discretized subgroup of $\mathrm{E}(2)$. A similar statement can also be found in symmetric MDP (Zinkevich & Balch, 2001), MDP homomorphism induced from symmetry group (Ravindran & Barto, 2004), and later work on symmetry in deep RL (van der Pol et al., 2020b; Wang et al., 2021).

***Theorem 2*** *For a GMDP, value iteration is an* $\mathrm{E}(d)$*-equivariant geometric message passing.*

*Proof.* We prove by constructing value iteration with For a more rigorous account on the relationship between dynamic programming (DP) and message passing on *non-geometric* MDPs, see (Dudzik & Veličković, 2022).

Notice that they satisfy the following equivariance conditions:

$$P_\theta : \mathcal{S} \times \mathcal{A} \times \mathcal{S} \to \mathbb{R}^+ : \quad P_\theta(\boldsymbol{s}_{t+1} \mid \boldsymbol{s}_t, \boldsymbol{a}_t) = P_\theta(\rho_\mathcal{S}(g) \cdot \boldsymbol{s}_{t+1} \mid \rho_\mathcal{S}(g) \cdot \boldsymbol{s}_t, \rho_\mathcal{A}(g) \cdot \boldsymbol{a}_t) \quad (23)$$

$$R_\theta : \mathcal{S} \times \mathcal{A} \to \mathbb{R} : \qquad\qquad R_\theta(\boldsymbol{s}_t, \boldsymbol{a}_t) = R_\theta(\rho_\mathcal{S}(g) \cdot \boldsymbol{s}_t, \rho_\mathcal{A}(g) \cdot \boldsymbol{a}_t) \quad (24)$$

$$Q_\theta : \mathcal{S} \times \mathcal{A} \to \mathbb{R} : \qquad\qquad Q_\theta(\boldsymbol{s}_t, \boldsymbol{a}_t) = Q_\theta(\rho_\mathcal{S}(g) \cdot \boldsymbol{s}_t, \rho_\mathcal{A}(g) \cdot \boldsymbol{a}_t) \quad (25)$$

$$V_\theta : \mathcal{S} \to \mathbb{R} : \qquad\qquad V_\theta(\boldsymbol{s}_t) = V_\theta(\rho_\mathcal{S}(g) \cdot \boldsymbol{s}_t) \quad (26)$$

$$(27)$$

We construct geometric message passing such that it uses *scalar* messages and features and resembles value iteration. The idea is visualized in Fig 9.

Then, we can use geometric message passing network to construct value iteration, which is to iteratively apply Bellman operators. We adopt the definition of geometric message passing based on

([Brandstetter et al., 2021](#)) as follows.

$$\tilde{\mathbf{m}}_{ij} = \phi_m\left(\tilde{\mathbf{f}}_i, \tilde{\mathbf{f}}_j, \tilde{\mathbf{a}}_{ij}\right) \tag{28}$$

$$\tilde{\mathbf{f}}_i' = \phi_f\left(\tilde{\mathbf{f}}_i, \sum_{j \in \mathcal{N}(i)} \tilde{\mathbf{m}}_{ij}, \tilde{\mathbf{a}}_i\right). \tag{29}$$

The tilde means they are steerable under $G$ transformations.

We want to construct value iteration:

$$Q(s,a) = R(s,a) + \gamma \sum_{s'} P(s'|s,a) V(s') \tag{30}$$

$$V'(s) = \sum_a \pi(a|s) Q(s,a) \tag{31}$$

To construct a *geometric graph*, we let vertices $\mathcal{V}$ be states $s$ and edges $\mathcal{E}$ be state-action transition $(s, a)$ labelled by $a$. For the geometric features on the graph, there are node features and edge features. Node features include maps/functions on the state space: $\mathcal{S} \to \mathbb{R}^D$, and edge features include functions on the state-action space $\mathcal{S} \times \mathcal{A} \to \mathbb{R}^D$.

For example, state value function $V : \mathcal{S} \to \mathbb{R}$ is (scalar) node feature, and $Q$-value function $Q_\theta : \mathcal{S} \times \mathcal{A} \to \mathbb{R}$ and reward function $R_\theta : \mathcal{S} \times \mathcal{A} \to \mathbb{R}$ are edge features. The message $\tilde{\mathbf{m}}_{ij}$ is thus a scalar for every edge: $\tilde{\mathbf{m}}_{ij} = \pi(a|s) Q(s,a)$, and $\tilde{\mathbf{f}}_i'$ is updated value function $\tilde{\mathbf{f}}_i' = V'(s)$. It is possible to extend value iteration to vector form as in Symmetric Value Iteration Network and Theorem 5.2 in ([Zhao et al., 2022b](#)), while we leave it for future work.

## D.2 Linear-Quadratic Control: Linearizing Geometric MDPs

Linear-Quadratic Regulator (LQR) is one of the most frequently used methods in optimal control [Tedrake (2023)](#). LQR is a computationally efficient method for solving problems with linear dynamics and quadratic costs. LQR has various noise robustness and optimally guarantees. Even if the dynamical system is nonlinear, linear-quadratic control methods have been used after iteratively linearizing the dynamics and quadratizing the cost.

Many of the problems where LQR is applied have symmetries. Recently has the control community began to study how symmetry can be used to increase the performance of classical control algorithms [Teng et al. (2023](#); [2022)](#); [Ghaffari et al. (2022)](#). [Hampsey et al. (2022b](#);[a)](#); [Cohen et al. (2020a)](#) specifically consider LQR on homogeneous spaces but do not establish the connection to steerable kernels.

We show how Euclidean symmetry inherently simplifies the linearized problem. We assume the dynamics and cost (reward) are **unknown** and need to be *learned*, thus equivariance constraints come in and *reduce the number of free parameters*.

## D.3 Theorem 3: Equivariance of Linearized Dynamics

We show the derivation of steerable kernel constraint in this subsection and further discuss the characteristics of steerable kernels in the next subsection.

***Theorem 3:*** *If a Geometric MDP has an infinitesimal $G$-action on the state-action space $\mathcal{S} \times \mathcal{A}$, the linearized dynamics is also $G$-equivariant: the matrix-value functions $A : \mathcal{S} \times \mathcal{A} \to \mathbb{R}^{d_{\mathcal{S}} \times d_{\mathcal{S}}}$ and $B : \mathcal{S} \times \mathcal{A} \to \mathbb{R}^{d_{\mathcal{S}} \times d_{\mathcal{A}}}$ satisfy $G$-steerable kernel constraints.*

Oftentimes, the problem of interest has continuous symmetry $G$ that acts on the space $\mathcal{S} \times \mathcal{A}$. We will assume that the Lie group action $G \times (\mathcal{S} \times \mathcal{A}) \to \mathcal{S} \times \mathcal{A}$ is continuous.

Under infinitesimal symmetry transformation $g \approx 1_G \in G$, let us suppose that the state and action space transform as

$$s \to \rho_{\mathcal{S}}(g) \cdot s, \quad a \to \rho_{\mathcal{A}}(g) \cdot a$$

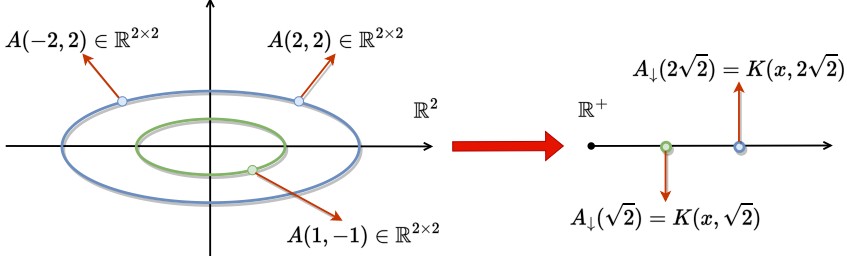

Figure 10: *Adopted from main text.* A schematic showing how a matrix-valued kernel $A : X \to \mathbb{R}^{2\times 2}$ is constrained by the SO(2)-steerable kernel constraints on a set of orbits $A(g \cdot p) = \rho_{\text{out}}(g)A(p)\rho_{\text{in}}(g^{-1})$. On each orbit (left), the constraints can be solved exactly: the matrices on same orbits (same colors) are related and have explicit parameterization given in Lang & Weiler (2020a). Thus, these matrices can be spanned on a basis (denoted by $K$) and live in a smaller "base" space $\mathcal{B} = X/G$ with a certain form $A_\downarrow : \mathcal{B} \to \mathbb{R}^{2\times 2}$.

where $\rho_\mathcal{S}$ and $\rho_\mathcal{A}$ are representations of the group $G$. Let $A$ and $B$ be the lineariziations of the dynamics $f$ at the point $p = (\boldsymbol{s}, \boldsymbol{a})$, with

$$A(p) = \frac{\partial f}{\partial s}|_p, \quad B(p) = \frac{\partial f}{\partial a}|_p$$

Under infinitesimal symmetry transformation $g \approx 1_G \in G$, the dynamics must satisfy,

$$\rho_\mathcal{S}(g) \cdot f(\boldsymbol{s}, \boldsymbol{a}) = f(\rho_\mathcal{S}(g) \cdot \boldsymbol{s}, \rho_\mathcal{A}(g) \cdot \boldsymbol{a}) \tag{32}$$

Because we assume the state and action space has *continuous* group action, we can apply Taylor expansion for continuous group actions and only keep the first order term. Let $g = 1_G + \delta g + \mathcal{O}(\delta g^2)$ be the expansion around the identity element, then

$$\rho_\mathcal{S}(g) = \rho_\mathcal{S}(1_G) + \rho_\mathcal{S}(\delta g) + \mathcal{O}(\delta g^2) = \mathbf{1}_{d_\mathcal{S}} + \rho_\mathcal{S}(\delta g) + \mathcal{O}(\delta g^2)$$
$$\rho_\mathcal{A}(g) = \rho_\mathcal{A}(1_G) + \rho_\mathcal{A}(\delta g) + \mathcal{O}(\delta g^2) = \mathbf{1}_{d_\mathcal{A}} + \rho_\mathcal{A}(\delta g) + \mathcal{O}(\delta g^2)$$

inserting into 32, and collecting terms of order $\mathcal{O}(\delta g)$, we have that,

$$\rho_\mathcal{S}(\delta g)\frac{\partial f}{\partial s}|_{\delta p} = \frac{\partial f}{\partial s}|_{\delta g \cdot p}\rho_\mathcal{S}(\delta g), \quad \rho_\mathcal{S}(\delta g)\frac{\partial f}{\partial a}|_p = \frac{\partial f}{\partial a}|_{\delta g \cdot p}\rho_\mathcal{S}(\delta g)$$

In general, we solve the non-linear problem by linearizing and iterating. Let us consider the linearized problem at point $p = (\boldsymbol{s}_0, \boldsymbol{a}_0)$. Assuming that the state and control do not change too drastically over a short period of time, we can approximate the true dynamics as

$$\boldsymbol{s}_{t+1} = A(p) \cdot \boldsymbol{s}_t + B(p) \cdot \boldsymbol{a}_t$$

where $A : \mathcal{S} \times \mathcal{A} \to \mathbb{R}^{d_\mathcal{S} \times d_\mathcal{S}}$ and $B : \mathcal{S} \times \mathcal{A} \to \mathbb{R}^{d_\mathcal{S} \times d_\mathcal{A}}$.

Now, linearizing the dynamics $f$ and using the symmetry constraint. The matrix valued functions $A(p)$ and $B(p)$ must satisfy the constraints

$$\forall g \in G, \quad A(g \cdot p) = \rho_\mathcal{S}(g)A(p)\rho_\mathcal{S}(g^{-1}), \quad B(g \cdot p) = \rho_\mathcal{S}(g)B(p)\rho_\mathcal{A}(g^{-1}) \tag{33}$$

so that $A$ is a $G$-steerable kernel with input representation $\rho_\mathcal{S}$ and output representation $\rho_\mathcal{S}$ and $B$ is a $G$-steerable kernel with input representation $\rho_\mathcal{A}$ and output representation $\rho_\mathcal{S}$ Cohen & Welling (2016a). The kernel constraints 33 relate $A(p)$ and $B(p)$ at different points. We use this constraint to understand why Euclidean symmetry is beneficial for decision-making: it *reduces* the number of free parameters and the dimensions of the solution space.

### D.4 CHARACTERISTICS OF $G$-STEERABLE KERNELS

Consider the $G$-steerable kernel constraints,

$$\forall g \in G, \quad A(g \cdot p) = \rho_\mathcal{S}(g)A(p)\rho_\mathcal{S}(g^{-1}), \quad B(g \cdot p) = \rho_\mathcal{S}(g)B(p)\rho_\mathcal{A}(g^{-1}) \tag{34}$$

To begin, note that this equation only relates $A$ and $B$ on points that can be related by $G$-transformation. As first observed in Weiler et al. (2018a), we can solve the constraints 34 over each orbit separately. Let us define the $G$-orbits of $\mathcal{S} \times \mathcal{A}$ as

$$O(x) = \{ \ y \ | \ \exists g \in G, \ y = g \cdot x \ \} \tag{35}$$

where $x \in \mathcal{S} \times \mathcal{A}$. Every $G$-orbit is a homogeneous space of $G$ Serre (2005). The set of $G$-orbits form a partition of the space $\mathcal{S} \times \mathcal{A}$. Let us define the equivalence relation $\sim$ as

$$x \sim y \implies \exists g \in G, \text{ such that } x = g \cdot y$$

so that $x \sim y$ only if $x$ and $y$ are related by symmetry transformation. We then define the quotient space

$$\mathcal{B} = (\mathcal{S} \times \mathcal{A})/\sim$$

where the space $\mathcal{B}$ consists of the space of all $G$-orbit representatives. Using a standard result in topology Husemöller (2013), there is then a canonical continuous projection map $\Pi : \mathcal{S} \times \mathcal{A} \to \mathcal{B}$ which projects each point in $\mathcal{S} \times \mathcal{A}$ to a canonically chosen orbit representative. Note that points related by $G$-action have the same projection and

$$\forall g \in G, \quad \Pi(g \cdot x) = \Pi(x)$$

holds for all $x \in \mathcal{S} \times \mathcal{A}$. We can decompose every point in the space $\mathcal{S} \times \mathcal{A}$ into an element of $\mathcal{B}$ and a element of a homogeneous space of $G$. Let $p \in \mathcal{S} \times \mathcal{A}$, we can always write

$$p = x_p \times p_{\downarrow}$$

where $p_{\downarrow} = \Pi(p) \in \mathcal{B}$ is the orbit representative of $p$ and $x_p$ is an element of $O(p_{\downarrow})$ which is a homogeneous space of $G$. Now, using this decomposition, we may write the constraints 33 as

$$\forall g \in G, \quad A(g \cdot x, p_{\downarrow}) = \rho_{\mathcal{S}}(g)A(x, p_{\downarrow})\rho_{\mathcal{S}}(g^{-1}), \quad B(g \cdot x, p_{\downarrow}) = \rho_{\mathcal{S}}(g)B(x, p_{\downarrow})\rho_{\mathcal{A}}(g^{-1}),$$

A complete solution to this constraint for any compact group $G$ was given in Lang & Weiler (2020a). Following Weiler et al. (2018a), we can simplify these kernel constraints based on the decomposition of $\rho_{\mathcal{S}}$ and $\rho_{\mathcal{A}}$ into irreducibles. Let $\hat{G}$ denote a representative set of $G$-irreducibles. Let us suppose that $\rho_{\mathcal{S}}$ and $\rho_{\mathcal{A}}$ decompose into irreducibles of $G$ as

$$(\rho_{\mathcal{S}}, V_{\mathcal{S}}) = \bigoplus_{\sigma \in \hat{G}} n_{\sigma}(\sigma, V_{\sigma}), \quad (\rho_{\mathcal{A}}, V_{\mathcal{A}}) = \bigoplus_{\sigma \in \hat{G}} q_{\sigma}(\sigma, V_{\sigma})$$

where $(\sigma, V_{\sigma})$ are the $G$-irreducibles and $n_{\sigma}$ and $q_{\sigma}$ count the multiplicity of the $(\sigma, V_{\sigma})$ irreducible in $\rho_{\mathcal{S}}$ and $\rho_{\mathcal{A}}$, respectively. The dimensions of each irreducible are related to the dimensions of the $\mathcal{S}$ and $\mathcal{A}$ manifolds via

$$d_{\mathcal{S}} = \sum_{\sigma \in \hat{G}} d_{\sigma} n_{\sigma}, \qquad d_{\mathcal{A}} = \sum_{\sigma \in \hat{G}} d_{\sigma} q_{\sigma}$$

Now, by definition of an reducible representation, there exists a $d_{\mathcal{S}} \times d_{\mathcal{S}}$ unitary matrix $U$ and a $d_{\mathcal{A}} \times d_{\mathcal{A}}$ unitary matrix $V$ such that we change basis and write

$$\forall g \in G, \quad \rho_{\mathcal{S}}(g) = U \begin{bmatrix} n_1\sigma_1(g) & 0 & 0 & ... & 0 & 0 \\ 0 & n_2\sigma_2(g) & 0 & ... & 0 & 0 \\ 0 & 0 & n_3\sigma_3(g) & ... & 0 & 0 \\ ... & ... & ... & ... & ... & ... \\ 0 & & 0 & ... & 0 & n_{|\hat{G}|}\sigma_{|\hat{G}|}(g) \end{bmatrix} U^{\dagger}$$

and

$$\forall g \in G, \quad \rho_{\mathcal{A}}(g) = V \begin{bmatrix} q_1\sigma_1(g) & 0 & 0 & ... & 0 & 0 \\ 0 & q_2\sigma_2(g) & 0 & ... & 0 & 0 \\ 0 & 0 & q_3\sigma_3(g) & ... & 0 & 0 \\ ... & ... & ... & ... & ... & ... \\ 0 & & 0 & ... & 0 & q_{|\hat{G}|}\sigma_{|\hat{G}|}(g) \end{bmatrix} V^{\dagger}$$

where the notation $b_i \sigma_i(g)$ denotes a block matrix with $b_i$ copies of the irreducible $(\sigma_i, V_i) \in \hat{G}$ on the diagonals,

$$\forall g \in G, \quad b_i \sigma_i(g) = \underbrace{\begin{bmatrix} \sigma_i(g) & 0 & 0 & ... & 0 & 0 \\ 0 & \sigma_i(g) & 0 & ... & 0 & 0 \\ 0 & 0 & \sigma_i(g) & ... & 0 & 0 \\ ... & ... & ... & ... & ... & ... \\ 0 & & 0 & ... & 0 & \sigma_i(g) \end{bmatrix}}_{b_i \text{ copies}}$$

We can write down the solution explicitly in this basis,

$$A(x, p_\downarrow) = U \begin{bmatrix} n_1 n_1 K_{11}^A(x, p_\downarrow) & n_1 n_2 K_{12}^A(x, p_\downarrow) & n_1 n_3 K_{13}^A(x, p_\downarrow) & ... & n_1 n_{|\hat{G}|} K_{1|\hat{G}|}^A(x, p_\downarrow) \\ n_2 n_1 K_{21}^A(x, p_\downarrow) & n_2 n_2 K_{22}^A(x, p_\downarrow) & n_2 n_3 K_{23}^A(x, p_\downarrow) & ... & n_2 n_{|\hat{G}|} K_{2|\hat{G}|}^A(x, p_\downarrow) \\ ... & ... & ... & ... & ... \\ n_{|\hat{G}|} n_1 K_{|\hat{G}|1}^A(x, p_\downarrow) & n_{|\hat{G}|} n_2 K_{|\hat{G}|2}^A(x, p_\downarrow) & n_{|\hat{G}|} n_3 K_{|\hat{G}|3}^A(x, p_\downarrow) & ... & n_{|\hat{G}|} n_{|\hat{G}|} K_{|\hat{G}||\hat{G}|}^A(x, p_\downarrow) \end{bmatrix} U^\dagger$$

and

$$B(x, p_\downarrow) = U \begin{bmatrix} n_1 q_1 K_{11}^B(x, p_\downarrow) & n_1 q_2 K_{12}^B(x, p_\downarrow) & n_1 q_3 K_{13}^B(x, p_\downarrow) & ... & n_1 q_{|\hat{G}|} K_{1|\hat{G}|}^B(x, p_\downarrow) \\ n_2 q_1 K_{21}^B(x, p_\downarrow) & n_2 q_2 K_{22}^B(x, p_\downarrow) & n_2 q_3 K_{23}^B(x, p_\downarrow) & ... & n_2 q_{|\hat{G}|} K_{2|\hat{G}|}^B(x, p_\downarrow) \\ ... & ... & ... & ... & ... \\ n_{|\hat{G}|} q_1 K_{|\hat{G}|1}^B(x, p_\downarrow) & n_{|\hat{G}|} q_2 K_{|\hat{G}|2}^B(x, p_\downarrow) & q_{|\hat{G}|} n_3 K_{|\hat{G}|3}^B(x, p_\downarrow) & ... & n_{|\hat{G}|} q_{|\hat{G}|} K_{|\hat{G}||\hat{G}|}^B(x, p_\downarrow) \end{bmatrix} V^\dagger$$

where $K_{ij}^A$ are the $A$-kernels with input representation $(\sigma_i, V_i)$ and output representation $(\sigma_j, V_j)$ and $K_{ij}^B$ are the $B$-kernels with input representation $(\sigma_i, V_i)$ and output representation $(\sigma_j, V_j)$. The notation $b_i c_j K_{ij}^C(x)$ denotes $b_i \times c_j$ independent copies of the $K_{ij}^C$ kernel,

$$b_i c_j K_{ij}^C(x, p_\downarrow) = \underbrace{\left.\begin{bmatrix} K_{ij}^C(x, p_\downarrow) & K_{ij}^C(x, p_\downarrow) & K_{ij}^C(x, p_\downarrow) & ... & K_{ij}^C(x, p_\downarrow) \\ K_{ij}^C(x, p_\downarrow) & K_{ij}^C(x, p_\downarrow) & K_{ij}^C(x, p_\downarrow) & ... & K_{ij}^C(x, p_\downarrow) \\ ... & ... & ... & ... & ... \\ K_{ij}^C(x, p_\downarrow) & K_{ij}^C(x, p_\downarrow) & K_{ij}^C(x, p_\downarrow) & ... & K_{ij}^C(x, p_\downarrow) \end{bmatrix}\right\} b_i \text{ copies}}_{c_j \text{ copies}}$$

Using Bra-Ket notation, let us define

$$K_{\sigma\rho}^{\tau ks}(x) = \sum_{i_\tau=1}^{d_\tau} \sum_{j_\sigma=1}^{d_\sigma} \sum_{i_\rho=1}^{d_\rho} |\sigma j_\sigma\rangle \underbrace{\langle s, \sigma j_\sigma | \tau i_\tau, \rho i_\rho\rangle}_{\text{Clebsch-Gordon}} \underbrace{Y_{\rho,k}^{i_\rho}(x)}_{\text{harmonics}} \langle \rho i_\rho|$$

Then, using the main result of Lang & Weiler (2020a), the matrices $K_{\sigma\rho}^{\tau ks}(x)$ form a basis for the space of $G$-steerable kernels with input representation $\rho$ and output representation $\sigma$. Then, the kernels $K^A$ and $K^B$ can be written in the form

$$K_{\sigma\rho}^A(x) = \sum_{\tau \in \hat{G}} \sum_{k=1}^{m_\rho} \sum_{s=1}^{m_\sigma} c_{\tau ks}^A(p_\downarrow) K_{\sigma\rho}^{\tau ks}(x)$$

$$K_{\sigma\rho}^A(x) = \sum_{\tau \in \hat{G}} \sum_{k=1}^{m_\rho} \sum_{s=1}^{m_\sigma} c_{\tau ks}^B(p_\downarrow) K_{\sigma\rho}^{\tau ks}(x)$$

where $c_{\tau ks}^A(p_\downarrow) : \mathcal{B} \to \text{Hom}_G[(\sigma, V_\sigma), (\sigma, V_\sigma)]$ and $c_{\tau ks}^B(p_\downarrow) : \mathcal{B} \to \text{Hom}_G[(\sigma, V_\sigma), (\sigma, V_\sigma)]$ are maps from the quotient space $\mathcal{B}$ into $(\sigma, V_\sigma)$-endomorphisms.

Thus, when $p_\downarrow \in \mathcal{B}$ is fixed, the total number of free parameters in the $A(x, p_\downarrow)$ matrix is

$$\dim A(x, p_\downarrow) = \sum_{\rho\sigma \in \hat{G}} n_\rho n_\sigma \sum_{\tau \in \hat{G}} m_\sigma m_\rho C_{\tau\rho}^\sigma \times \dim \text{Hom}_G[(\sigma, V_\sigma), (\sigma, V_\sigma)] \leq 4 \sum_{\rho\tau\sigma \in \hat{G}} n_\rho n_\sigma C_{\tau\rho}^\sigma m_\sigma m_\rho$$

Similarly, at fixed $p_\downarrow \in \mathcal{B}$, the total number of free parameters in $B(x, p_\downarrow)$

$$\dim B(x, p_\downarrow) = \sum_{\rho\sigma\in\hat{G}} n_\rho q_\sigma \sum_{\tau\in\hat{G}} C^\sigma_{\tau\rho} m_\sigma m_\rho \times \dim \mathrm{Hom}_G[(\sigma, V_\sigma), (\sigma, V_\sigma)] \leq 4 \sum_{\rho\tau\sigma\in\hat{G}} n_\rho q_\sigma C^\sigma_{\tau\rho} m_\sigma m_\rho$$

Note that for fixed $p_\downarrow$,

$$\dim A(x, p_\downarrow) \leq d_\mathcal{S}^2, \quad \dim B(x, p_\downarrow) \leq d_\mathcal{S} d_\mathcal{A},$$

always hold. To summarize, symmetry constraints force the LQR matrices $A$ and $B$ to take the form

$$A : \mathcal{B} \to \mathbb{R}^{d_\mathcal{S}\times d_\mathcal{S}}, \quad B : \mathcal{B} \to \mathbb{R}^{d_\mathcal{S}\times d_\mathcal{A}}$$

Furthermore, the output matrices take the form of a $G$-steerable kernel Lang & Weiler (2020a) and are parameterized by only a small number of parameters. This should be contrasted with the non-equivarient case where

$$A : \mathcal{S} \times \mathcal{A} \to \mathbb{R}^{d_\mathcal{S}\times d_\mathcal{S}}, \quad B : \mathcal{S} \times \mathcal{A} \to \mathbb{R}^{d_\mathcal{S}\times d_\mathcal{A}}$$

The dimension of the space $\mathcal{B}$ can be significantly less than that of $\mathcal{S} \times \mathcal{A}$. Symmetry constraints thus highly restricts the allowed form of LQR.

## D.5 THEOREM 4: SYMMETRY IN SOLUTIONS OF LQR

***Theorem 4:*** *The LQR feedback matrix in $a_t^\star = -K(p)s_t$ and value matrix in $V = s_t^\top P(p)s_t$ are $G$-steerable kernels, or matrix-valued functions: $K : \mathcal{S} \times \mathcal{A} \to \mathbb{R}^{d_\mathcal{A}\times d_\mathcal{S}}$ and $P : \mathcal{S} \times \mathcal{A} \to \mathbb{R}^{d_\mathcal{S}\times d_\mathcal{S}}$.*

*Proof.* The solution of LQR is derived from Bellman equation. The results, or optimal value function $V_t(s) = s^\top P_t s$ and optimal policy function (feedback control law) $a^\star = -K_t s$, are given by the discrete algebraic Riccati equation (DARE).

$$V_t(s) = s^\top P_t s, \quad a^\star = -K_t s, \tag{36}$$

where

$$P_t = Q + A^\top P_{t+1} A - A^\top P_{t+1} B \left(R + B^\top P_{t+1} B\right)^{-1} B^\top P_{t+1} A \tag{37}$$

$$K_t = \left(R + B^\top P_{t+1} B\right)^{-1} B^\top P_{t+1} A. \tag{38}$$

The goal is to prove that $P_t$ and $K_t$ are $G$-steerable kernels.

If we iteratively linearize the problem, similar to the linearized dynamics, we quadratize the cost function around a point $p$. The quadratic cost also depends on the point $p$ and is given as follows.

$$s_{t+1} = A(p) \cdot s_t + B(p) \cdot a_t, \quad A : \mathcal{S} \times \mathcal{A} \to \mathbb{R}^{d_\mathcal{S}\times d_\mathcal{S}}, \quad B : \mathcal{S} \times \mathcal{A} \to \mathbb{R}^{d_\mathcal{S}\times d_\mathcal{A}} \tag{39}$$

$$c(s, a) = s^\top Q(p) s + a^\top R(p) a, \quad Q : \mathcal{S} \times \mathcal{A} \to \mathbb{R}^{d_\mathcal{S}\times d_\mathcal{S}}, \quad R : \mathcal{S} \times \mathcal{A} \to \mathbb{R}^{d_\mathcal{A}\times d_\mathcal{A}} \tag{40}$$

Analogously, $Q$ and $R$ are also $G$-steerable kernels by assuming the scalar function $c(s, a)$ is $G$-invariant:

$$\forall g \in G, \quad Q(g \cdot p) = \rho_\mathcal{S}(g)Q(p)\rho_\mathcal{S}(g^{-1}), \quad R(g \cdot p) = \rho_\mathcal{A}(g)R(p)\rho_\mathcal{A}(g^{-1}) \tag{41}$$

We prove by induction. For simplicity, we denote $A_p := A(p)$ and similarly for $B_p, Q_p, R_p$. We start from $P_T = Q_p$. By the property of $G$-steerable kernel, it is $Q_p = \rho_\mathcal{S}(g)Q_{g\cdot p}\rho_\mathcal{S}(g^{-1})$, thus $P_T$ is a steerable kernel.

By induction, we assume $P_{t+1}$ is steerable kernel: $P_{t+1}(p) = \rho_\mathcal{S}(g)P_{t+1}(g \cdot p)\rho_\mathcal{S}(g^{-1})$. We show how each component is transformed under group transformation.

$$P_t(p) = \underbrace{Q_p}_{(1)} + \underbrace{A_p^\top P_{t+1}(p)A_p}_{(2)} - \underbrace{A_p^\top P_{t+1}(p)B_p}_{(3)} \underbrace{\left(R_p + B_p^\top P_{t+1}(p)B_p\right)^{-1}}_{(4)} \underbrace{B_p^\top P_{t+1}(p)A_p}_{(5)} \tag{42}$$

For (1), by the property of $G$-steerable kernel, it is $Q_p = \rho_\mathcal{S}(g)Q_{g\cdot p}\rho_\mathcal{S}(g^{-1})$.

Table 3: Equivariant dimension reduction for linearized dynamics. This table highlights the reduced dimensions of spaces of kernels. $\mathcal{X}(\mathcal{M})$ denotes the dimension of signals living on the manifold $\mathcal{M}$.

| Task | $\mathcal{S}$ | $\mathcal{A}$ | $G$ | $\rho_{\mathcal{S}}$ | $\rho_{\mathcal{A}}$ | $\mathcal{X}(\mathcal{S} \times \mathcal{A})$ | $\mathcal{X}(\mathcal{B})$ |
|---|---|---|---|---|---|---|---|
| Free Particle in 2D | $\mathbb{R}^2 \times \mathbb{R}^2$ | $\mathbb{R}^2$ | SO(2) | $\rho_{std}$ | $\rho_{std}$ | $\mathbb{R}^{16}$ | $\mathbb{R}^+ \times \mathbb{R}^{14}$ |
| Reacher (in 2D) | $S^1 \times S^1 \times \mathbb{R}^2$ | $\mathbb{R}^2$ | SO(2) | $\rho_{std} \oplus \rho_{triv}$ | $\rho_{triv}$ | $S^1 \times S^1 \times \mathbb{R}^{10}$ | $S^1 \times \mathbb{R}^{10}$ |
| Single Free Particle in 3D | $\mathbb{R}^3 \times \mathbb{R}^3$ | $\mathbb{R}^3$ | SO(3) | $\rho_{std}$ | $\rho_{std}$ | $\mathbb{R}^{32}$ | $(\mathbb{R}^+)^2 \times \mathbb{R}^{10}$ |
| $N$-Free Particles in 3D | $\mathbb{R}^{3N} \times \mathbb{R}^{3N}$ | $\mathbb{R}^{3N}$ | SO(3) | $\rho_{std}$ | $\rho_{std}$ | $\mathbb{R}^{32N}$ | $(\mathbb{R}^+)^2 \times \mathbb{R}^{10N}$ |

For (2), we have

$$A_p^\top P_{t+1}(p) A_p = \left(\rho_{\mathcal{S}}(g) A_{g \cdot p} \rho_{\mathcal{S}}\left(g^{-1}\right)\right)^\top \left(\rho_{\mathcal{S}}(g) P_t(g \cdot p) \rho_{\mathcal{S}}(g^{-1})\right) \left(\rho_{\mathcal{S}}(g) A_{g \cdot p} \rho_{\mathcal{S}}\left(g^{-1}\right)\right) \quad (43)$$

$$= \rho_{\mathcal{S}}(g) A_{g \cdot p}^\top \rho_{\mathcal{S}}\left(g^{-1}\right) \rho_{\mathcal{S}}(g) P_t(g \cdot p \rho_{\mathcal{S}}(g^{-1})) \rho_{\mathcal{S}}(g) A_{g \cdot p} \rho_{\mathcal{S}}\left(g^{-1}\right) \quad (44)$$

$$= \rho_{\mathcal{S}}(g) \left(A_{g \cdot p}^\top P_t(g \cdot p) A_{g \cdot p}\right) \rho_{\mathcal{S}}\left(g^{-1}\right). \quad (45)$$

We can similarly show for the rest:

$$(3) = \rho_{\mathcal{S}}(g) \left(A_{g \cdot p}^\top P_{t+1}(g \cdot p) B_{g \cdot p}\right) \rho_{\mathcal{A}}(g^{-1}) \quad (46)$$

$$(4)^{-1} = \rho_{\mathcal{A}}(g) \left(R_{g \cdot p} + B_{g \cdot p}^\top P_{t+1}(g \cdot p) B_{g \cdot p}\right) \rho_{\mathcal{A}}(g^{-1}) \quad (47)$$

$$(5) = \rho_{\mathcal{A}}(g) \left(B_{g \cdot p}^\top P_{t+1}(g \cdot p) A_{g \cdot p}\right) \rho_{\mathcal{S}}(g^{-1}) \quad (48)$$

Thus, the multiplication (3)(4)(5) transforms under input representation $\rho_{\mathcal{S}}(g)$ and output representation $\rho_{\mathcal{S}}(g^{-1})$. Analogously, all terms (1), (2), and (3)(4)(5) transforms under input representation $\rho_{\mathcal{S}}(g)$ and output representation $\rho_{\mathcal{S}}(g^{-1})$, same for the sum. By moving the terms, we obtain $P_t(p) = \rho_{\mathcal{S}}(g) P_t(g \cdot p) \rho_{\mathcal{S}}(g^{-1})$ that $P_t$ is a steerable kernel.

### D.6 ILLUSTRATION AND EXAMPLES

We work out a few examples of how symmetry can reduce the dimensionality of the LQR problem. We specifically consider two simple problems: a free particle moving in the plane and a free particle moving in space. The agent can control an applied force and tries to steer the particle to the origin.

**Examples.** In Table 3, we show the dimensions of the spaces of $G$-steerable kernels for each task. $\mathcal{X}(\mathcal{S} \times \mathcal{A})$ refers to the space *without* equivariant constraints, while $\mathcal{X}(\mathcal{B})$ denotes the space with *G-steerable equivariant constraints*. We can see that the spaces of equivariant version are hugely reduced because of (1) smaller base space, as visualized in Figure **??**, and (2) constrained output matrix $\mathbb{R}^{d \times d}$ with less free parameters.

In the empirical results, we show for multiple free particles ($N$-ball `PointMass` 3D), which generalize the results of single balls. Compared to `Reacher` that has two connected links by a joint, multiple free particles can have much better saving. Note that for `Reacher`, in implementation we convert the angles to unit vectors:

$$\left(\theta_1, \theta_2, \dot{\theta}_1, \dot{\theta}_2, x_g - x_f, y_g - y_f\right) \Rightarrow \left(\cos \theta_1, \sin \theta_1, \cos \theta_2, \sin \theta_2, \dot{\theta}_1, \dot{\theta}_2, x_g - x_f, y_g - y_f\right). \quad (49)$$

### D.6.1 SINGLE PARTICLE CONTROL IN THE PLANE

We work out the single particle control problem in the plane. The goal is to move a particle to origin. The state space is the particle position, represented as a two-vector $\vec{p}$, and the particle velocity represented as a two-vector $\vec{v}$. The action space consists of the applied force $\hat{F}$, which is also a two-vector. The state space and action space are thus given by

$$\mathcal{S} = \mathbb{R}^2 \times \mathbb{R}^2, \qquad \mathcal{A} = \mathbb{R}^2$$

The group $SO(2)$ acts on the state and action space. Specifically, under a rotation $R \in SO(2)$, the state and action transform as

$$\text{State Transform: } (\vec{p}, \vec{v}) \rightarrow (R\vec{p}, R\vec{v})$$

$$\text{Action Transform: } \vec{F} \rightarrow R\vec{F}$$

The state space transforms in $\rho_{\mathcal{S}} = \rho_1 \oplus \rho_1$ and the action space transform in $\rho_{\mathcal{A}} = \rho_1$. The set of orbits are then given by states and actions where the angles between the vectors $\vec{p}, \vec{v}$ and $\vec{F}$ are fixed.

The base space is given by $\mathcal{B} = \mathbb{R}^+ \times \mathbb{R}^4$. Using Proposition E.6. in Lang & Weiler (2020a), a basis for the steerable kernels of input type $(\rho_1, V_1)$ and output type $(\rho_1, V_1)$ is given by

$$K_{11}(x) = c_1 \begin{bmatrix} 1 & 0 \\ 0 & 1 \end{bmatrix} + c_2 \begin{bmatrix} 0 & -1 \\ 1 & 0 \end{bmatrix} + c_3 \begin{bmatrix} \cos(2x) & \sin(2x) \\ \sin(2x) & -\cos(2x) \end{bmatrix} + c_4 \begin{bmatrix} -\sin(2x) & \cos(2x) \\ \cos(2x) & -\sin(2x) \end{bmatrix}$$

where each $c_j \in \mathbb{R}$. This result was first derived in Weiler et al. (2018a). In more compact notation, the matrix $K_{11} : S^1 \rightarrow \mathbb{R}^{2 \times 2}$ can be expanded as

$$K_{11}(x) = \begin{bmatrix} c_1 + c_3 \cos(2x) - c_4 \sin(2x), & -c_1 + c_3 \sin(2x) - c_4 \cos(2x) \\ c_2 + c_3 \sin(2x) + c_4 \cos(2x), & c_1 + c_3 \sin(2x) - c_4 \sin(2x) \end{bmatrix}$$

where each $c_i \in \mathbb{R}$. Then, using the form of the LQR matrices, we can write

$$A(p_\downarrow, x) = \begin{bmatrix} K_A^{(1,1)}(p_\downarrow, x), & K_A^{(1,2)}(p_\downarrow, x) \\ K_A^{(2,1)}(p_\downarrow, x), & K_A^{(2,2)}(p_\downarrow, x) \end{bmatrix}, \quad B(p_\downarrow, x) = \begin{bmatrix} K_B^{(1,1)}(p_\downarrow, x) \\ K_B^{(2,1)}(p_\downarrow, x) \end{bmatrix}$$

where each $K_A^{(i,j)}$ and $K_B^{(k,l)}$ take the form, $X \in \{A, B\}$,

$$K_X^{(i,j)}(p_\downarrow, x) = \begin{bmatrix} c_{1,X}^{(i,j)}(p_\downarrow) + c_{3,X}^{(i,j)}(p_\downarrow) \cos(2x) - c_{4,X}^{(i,j)}(p_\downarrow) \sin(2x), & -c_{1,X}^{(i,j)}(p_\downarrow) + c_{3,X}^{(i,j)}(p_\downarrow) \sin(2x) - c_{4,X}^{(i,j)}(p_\downarrow) \cos(2x) \\ c_{2,X}^{(i,j)}(p_\downarrow) + c_{3,X}^{(i,j)}(p_\downarrow) \sin(2x) + c_{4,X}^{(i,j)}(p_\downarrow) \cos(2x), & c_{1,X}^{(i,j)}(p_\downarrow) + c_{3,X}^{(i,j)}(p_\downarrow) \sin(2x) - c_{4,X}^{(i,j)}(p_\downarrow) \sin(2x) \end{bmatrix}$$

the coefficients $c_{kA}^{(i,j)} : \mathbb{R}^+ \times \mathbb{R}^4 \rightarrow \mathbb{R}$ and $c_{kB}^{(i,j)} : \mathbb{R}^+ \times \mathbb{R}^4 \rightarrow \mathbb{R}$ are not constrained by symmetry and can be parameterized by a neural network. If we amalgamate each of the coefficients $c_{kA}^{(i,j)}$ and $c_{kB}^{(i,j)}$ into a single vector output $C$, the system dynamics can be learned by specifying

$$C : \mathbb{R}^+ \times \mathbb{R}^4 \rightarrow \mathbb{R}^{16+8}$$

This should be contrasted with the non-equivarient case, where one needs to learn

$$A : \mathbb{R}^6 \rightarrow \mathbb{R}^{16} \text{ and } B : \mathbb{R}^6 \rightarrow \mathbb{R}^8$$

The dimensional reduction in the equivarient vs non-equivarient case is infinite. This is analogous to Wang et al. (2020) where equivarient methods can outperform non-equivarient methods by essentially an essentially infinite margin. In practice, due to discritization, this infinite gain is reduced to some large finite number.

### D.6.2 SINGLE PARTICLE CONTROL IN SPACE

Let us consider the analogous single particle control problem in three-dimensional space. The state space is the particle position, represented as a three-vector $\vec{p}$, and the particle velocity represented as a three-vector $\vec{v}$. The action space consists of the applied force $\hat{F}$, which is also a three-vector. The state space and action space are thus given by

$$\mathcal{S} = \mathbb{R}^3 \times \mathbb{R}^3, \qquad \mathcal{A} = \mathbb{R}^3$$

The group $SO(3)$ acts on the state and action space. Specifically, under a rotation $R \in SO(3)$, the state and action transform as

$$\text{State Transform: } (\vec{p}, \vec{v}) \rightarrow (R\vec{p}, R\vec{v})$$

$$\text{Action Transform: } \vec{F} \rightarrow R\vec{F}$$

Position and Velocity are both vectors the state space transforms in $\rho_{\mathcal{S}} = \rho_1 \oplus \rho_1$. Force is a vector quantity and the action space transform in $\rho_{\mathcal{A}} = \rho_1$. The set of $SO(3)$-orbits are then given by states

and actions where the angles between the vectors $\vec{p}$, $\vec{v}$ and $\vec{F}$ are fixed. The base space is given by $\mathcal{B} = \mathbb{R}^+ \times \mathbb{R}^6$.

A complete characterization of $SO(3)$-kernels was first derived in Weiler et al. (2018b). In the basis that diogonalizes the representation, the vectored kernel matrix for a $SO(3)$-steerable kernel of input type $(D^1, W^1)$ and $(D^1, W^1)$ can be written as

$$\text{Vec}(K_{11}(x)) = \begin{bmatrix} \Phi_0(||x||)Y_0(\frac{x}{||x||}) \\ \Phi_1(||x||)Y_1(\frac{x}{||x||}) \\ \Phi_2(||x||)Y_2(\frac{x}{||x||}) \end{bmatrix}$$

where each $Y_\ell : S^2 \to \mathbb{R}^{(2\ell+1)}$ are spherical harmonics in vector form. Each $\Phi_i(||x||) : \mathbb{R}^+ \to \mathbb{R}$ are a set of radial functions. Unvectorizing, we have that

$$K_{11}(x) = \begin{bmatrix} \Phi_0(||x||)Y_0^0(\frac{x}{||x||}) & \Phi_1(||x||)Y_1^1(\frac{x}{||x||}) & \Phi_2(||x||)Y_2^2(\frac{x}{||x||}) \\ \Phi_1(||x||)Y_1^{-1}(\frac{x}{||x||}) & \Phi_1(||x||)Y_1^0(\frac{x}{||x||}) & \Phi_2(||x||)Y_2^1(\frac{x}{||x||}) \\ \Phi_1(||x||)Y_2^{-2}(\frac{x}{||x||}) & \Phi_2(||x||)Y_2^{-1}(\frac{x}{||x||}) & \Phi_2(||x||)Y_2^0(\frac{x}{||x||}) \end{bmatrix}$$

Now, using the results of $G$-steerable kernel constraints, the most general LQR matrices can be written in the form

$$A(p_\downarrow, x) = \begin{bmatrix} K_A^{(1,1)}(p_\downarrow, x), & K_A^{(1,2)}(p_\downarrow, x) \\ K_A^{(2,1)}(p_\downarrow, x), & K_A^{(2,2)}(p_\downarrow, x) \end{bmatrix}, \quad B(p_\downarrow, x) = \begin{bmatrix} K_B^{(1,1)}(p_\downarrow, x) \\ K_B^{(2,1)}(p_\downarrow, x) \end{bmatrix}$$

where each $K_A^{(i,j)}$ and $K_B^{(k,l)}$ take the form, $X \in \{A, B\}$,

$$K_X^{ij}(x) = \begin{bmatrix} \Phi_0^{ij}(||x||)Y_0^0(\frac{x}{||x||}) & \Phi_1^{ij}(||x||)Y_1^1(\frac{x}{||x||}) & \Phi_2^{ij}(||x||)Y_2^0(\frac{x}{||x||}) \\ \Phi_1^{ij}(||x||)Y_1^{-1}(\frac{x}{||x||}) & \Phi_2^{ij}(||x||)Y_{-2}^{-2}(\frac{x}{||x||}) & \Phi_2^{ij}(||x||)Y_2^1(\frac{x}{||x||}) \\ \Phi_1^{ij}(||x||)Y_3^1(\frac{x}{||x||}) & \Phi_2^{ij}(||x||)Y_2^{-1}(\frac{x}{||x||}) & \Phi_2^{ij}(||x||)Y_2^2(\frac{x}{||x||}) \end{bmatrix}$$

the coefficients $\Phi_{kA}^{(i,j)} : \mathbb{R}^+ \times \mathbb{R}^6 \to \mathbb{R}$ and $\Phi_{kB}^{(i,j)} : \mathbb{R}^+ \times \mathbb{R}^6 \to \mathbb{R}$ are not constrained by symmetry and can be parameterized by a neural network. If we amalgamate each of the coefficients $\Phi_{kB}^{(i,j)}$ and $\Phi_{kB}^{(i,j)}$ into a single vector output $\Phi$, the system dynamics can be learned by specifying

$$\Phi : \mathbb{R}^+ \times \mathbb{R}^6 \to \mathbb{R}^{12+6}$$

This should be contrasted with the non-equivarient case where one needs to learn a function from $\mathbb{R}^3 \times \mathbb{R}^6 \to \mathbb{R}^{12+6}$. By utilizing symmetry, we are able to significantly reduce the domain of the function spaces.

## E ALGORITHM DESIGN

We elaborate on the algorithm design in this section.

**Invariance of `return`.** We compute the expected return of sampled trajectories, and study how it is transformed:

$$\text{return}(\tau) = \mathbb{E}_\tau \left[ \gamma^H Q_\theta(\boldsymbol{s}_H, \boldsymbol{a}_H) + \sum_{t=0}^{H-1} \gamma^t R_\theta(\boldsymbol{s}_t, \boldsymbol{a}_t) \right] = \mathbb{E}_\tau [U(\boldsymbol{s}_{1:H}, \boldsymbol{a}_{1:H-1})] \tag{50}$$

$$\text{return}(g \cdot \tau) = \mathbb{E}_{g \cdot \tau} \left[ \gamma^H \rho_0(g) \cdot Q_\theta(g \cdot \boldsymbol{s}_H, g \cdot \boldsymbol{a}_H) + \sum_{t=0}^{H-1} \gamma^t \rho_0(g) \cdot R_\theta(g \cdot \boldsymbol{s}_t, g \cdot \boldsymbol{a}_t) \right] \tag{51}$$

$$= \int_{g \in G} \rho_0(g) dg \cdot \mathbb{E}_\tau [U(g \cdot \boldsymbol{s}_{1:H}, g \cdot \boldsymbol{a}_{1:H-1})] \tag{52}$$

$$= \mathbf{1} \cdot \mathbb{E}_\tau [U(\boldsymbol{s}_{1:H}, \boldsymbol{a}_{1:H-1})] = \text{return}(\tau) \tag{53}$$

In Equation 51, we use $\rho_0(g) = \mathbf{1}$ to denote that the output is not transformed, so we may extract the term out. In Equation 52, $dg$ is a Haar measure that absorbs the normalization factor, and we can extract the term from expectation. Equation 53 uses the invariance of $Q_\theta$ and $R_\theta$. In other words, the return under the $G$-orbit of trajectories is the same, thus `return` is $G$-invariant.

**Equivariance of $G$-`sample`.** In Model Predictive Path Integral (MPPI) (Williams et al., 2017b), we sample $N$ actions from a random Gaussian distribution $\mathcal{N}(\mu, \sigma^2 I)$, denoted as $\mathbb{A} = \{a_i\}_{i=1}^N$. However, since $\mu$ and $\sigma$ are not state-dependent, CEM/MPPI does not satisfy the condition of equivariance, which requires that rotating the input results in a rotated output. To address this, we propose a solution - augmenting the action sampling by transforming with all elements in the group $G$: $G\mathbb{A} = \{g \cdot a_i \mid g \in G\}_{i=1}^N$. This approach ensures that our method can handle different orientations and maintain the property of equivariance.

To validate our approach, we first demonstrate the equivariance condition mathematically. We assume that $(s_0, a_0)$ gives the maximum value $a_0 = \arg\max_{a \in G\mathbb{A}} Q(s_0, a)$. If we consider $g \cdot a_0 = g \cdot \arg\max_{a \in G\mathbb{A}} Q(s_0, a) = \arg\max_{a \in G\mathbb{A}} Q(g \cdot s_0, a)$, it implies that if we rotate the state to $g \cdot s_0$, we expect $g \cdot a_0$ to still provide the maximum $Q$-value so that $\arg\max$ can select it. The proof is validated using the invariance of $Q$, $Q(g \cdot s, g \cdot a) = Q(s, a)$. Hence, $a_0' = \arg\max_{a \in G\mathbb{A}} Q(g \cdot s_0, a) = \arg\max_{a \in G\mathbb{A}} Q(s_0, g^{-1} \cdot a)$. By comparing these two equations, we find that $a_0' = g \cdot a_0$.

Note that when not augmenting $\mathbb{A}$, it is not guaranteed that $g \cdot a_0$ exists in $\mathbb{A}$. However, when the number of samples approaches infinity, $g \cdot a_0$ can get close to some element in $\mathbb{A}$.

The proof can be directly applied to multiple steps, as `return` is also $G$-invariant.

# F IMPLEMENTATION DETAILS AND ADDITIONAL EVALUATION

## F.1 IMPLEMENTATION DETAILS: EQUIVARIANT TD-MPC

We mostly follow the implementation of TD-MPC (Hansen et al., 2022). The training of TD-MPC is end-to-end, i.e., it produces trajectories with a learned dynamics and reward model and predicts the values and optimal actions for those states. It closely resembles MuZero (Schrittwieser et al., 2019) while uses MPPI (Model Predictive Path Integral (Williams et al., 2015; 2017b)) for continuous actions instead of MCTS (Monte-Carlo tree search) for discrete actions. It inherits the drawbacks from MuZero - the dynamics model is trained only from reward signals and may collapse or experience instability on sparse-reward tasks. This is also the case for the tasks we use: `PointMass` and `Reacher` and their variants, where the objectives are to reach a goal position.

## F.2 EXPERIMENTAL DETAILS

We implement $G$-equivariant MLP using `escnn` (Weiler & Cesa, 2021) for policy, value, transition, and reward network, with 2D and 3D discrete groups. For all MLPs, we use two layers with $512$ hidden units. The hidden dimension is set to be $48$ for non-equivariant version, and the equivariant version is to keep the same number of free parameters, or `sqrt` strategy.

For example, for $D_8$ group, `sqrt` strategy (to keep same free parameters) has number of hidden units divided by $\sqrt{|D_8|} = \sqrt{16} = 4$. The other strategy is to make equivariant networks' input and output be compatible with non-equivariant ones: *linear* strategy, which keeps same input/output dimensions (number of hidden units divided by $|D_8| = 16$).

The hidden space uses *regular* representation, which is common for discrete equivariant network (Cohen & Welling, 2016c; Weiler & Cesa, 2021; Zhao et al., 2022b).

## F.3 ADDITIONAL RESULTS

**Ablation on model-based vs. model-free ("planning-free").** We ablate the use of planning component in equivariant version of TD-MPC, which is to justify why we aim to build model-based version of equivariant RL algorithm over model-free counterparts. The results are shown in Figure 11. On both `Reacher` Easy and Hard, with planning, the performance is much better.

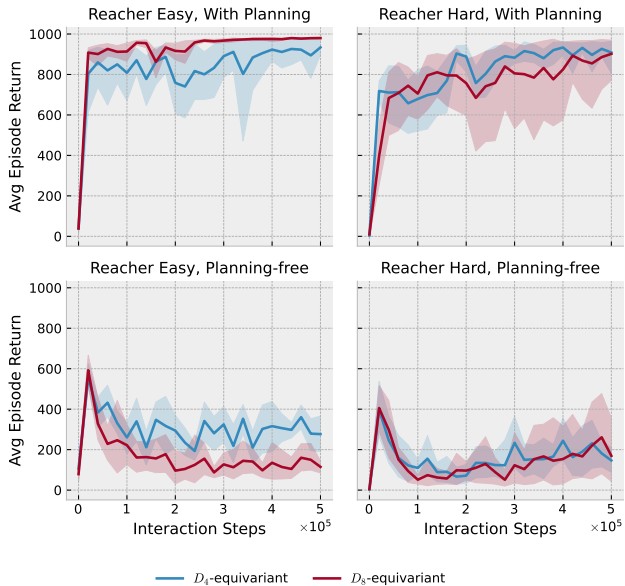

Figure 11: Ablation study on planning component.

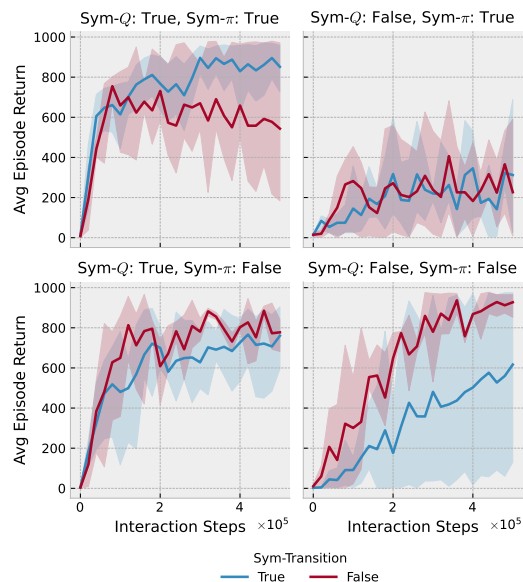

Figure 12: Ablation study on equivariant components, using `Reacher` Hard with $D_8$ symmetry group.

**Ablation on equivariant components.** Recall that we have several equivariant components in equivariant TD-MPC:

$$f_\theta : \mathcal{S} \times \mathcal{A} \to \mathcal{S} : \qquad \rho_\mathcal{S}(g) \cdot f_\theta(s_t, a_t) = f_\theta(\rho_\mathcal{S}(g) \cdot s_t, \rho_\mathcal{A}(g) \cdot a_t) \qquad (54)$$

$$R_\theta : \mathcal{S} \times \mathcal{A} \to \mathbb{R} : \qquad R_\theta(s_t, a_t) = R_\theta(\rho_\mathcal{S}(g) \cdot s_t, \rho_\mathcal{A}(g) \cdot a_t) \qquad (55)$$

$$Q_\theta : \mathcal{S} \times \mathcal{A} \to \mathbb{R} : \qquad Q_\theta(s_t, a_t) = Q_\theta(\rho_\mathcal{S}(g) \cdot s_t, \rho_\mathcal{A}(g) \cdot a_t) \qquad (56)$$

$$\pi_\theta : \mathcal{S} \to \mathcal{A} : \qquad \rho_\mathcal{A}(g) \cdot \pi_\theta(\cdot \mid s_t) = \pi_\theta(\cdot \mid \rho_\mathcal{S}(g) \cdot s_t) \qquad (57)$$

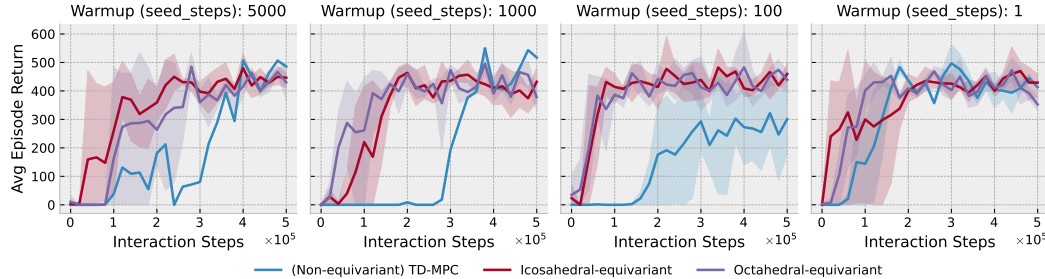

Figure 13: Ablation study on number of warmup episodes on `PointMass` *3D with small target*.

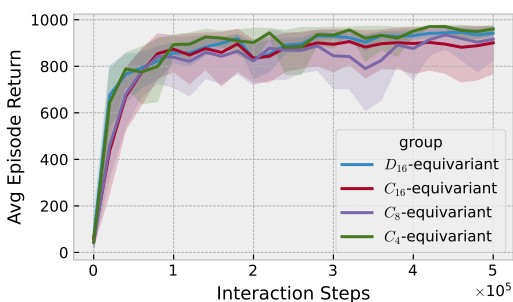

Figure 14: Ablation study on symmetry group on `Reacher` Hard.

We experiment to enable and disable each of them: (1) transition network: dynamics $f$ and reward $R$, (2) value network: $Q$, and (3) policy network $\pi$. Note that to make equivariant and non-equivariant components compatible, we need to make sure the input and output dimensions match.

We show the results on `Reacher` Hard with $D_8$ symmetry group in Fig 12. Instead of using `sqrt` strategy (to keep same free parameters, number of hidden units divided by $\sqrt{|D_8|} = \sqrt{16} = 4$) on specifying the number of hidden units, we use *linear* strategy that keeps same input/output dimensions (number of hidden units divided by $|D_8| = 16$). Thus, the performance of fully non-equivariant model and fully equivariant model are not directly comparable, because the number of free parameters in fully equivariant one is much smaller.

The results show the relative importance of value, policy, and transition. It shows the most important equivariant component is $Q$-value network. It is reasonable because it has been used intensively in predicting into the future, where generalization and training efficiency are very important and benefit from equivariance.

**Hyperparameter of amount of warmup.** We experiment different number of warmup episodes, called `seed steps` in TD-MPC hyperparameter. We find this is a critical hyperparameter for (non-equivariant) TD-MPC. One possible reason is that TD-MPC highly relies on joint training and may collapse when the transition model is stuck at some local minima. This warmup hyperparameter controls how many episodes TD-MPC collects before starting actual training.

We test using different numbers on `PointMass` *3D with small target*. The results are shown in Figure 13, which demonstrate that our equivariant version is robust under all choices of warmup episodes, even with little to none warmup. The non-equivariant TD-MPC is very sensitive to the choice of warmup number.

**Ablation on symmetry groups.** We also do ablation study on the choice of discrete subgroups. We run experiments on `Reacher` Hard to compare 2D discrete rotation/dihedral groups: $C_4, C_8, C_{16}, D_{16}$, using 1 warmup episode.

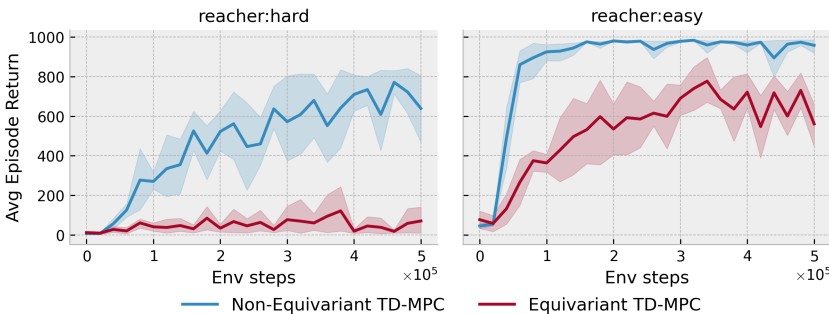

Figure 15: Results for global reference frame on `Reacher`.

The results are shown in Fig 14. We find using groups larger than $C_8$ does not bring additional improvement on this specific task, `Reacher` Hard. In the main paper, we thus use $D_4, D_8$ to balance the performance and computation time and memory use.

**Comparing reference frames and state features.** This experiment studies the balance between reference frames and the choice of state features. In the theory section, we emphasize that kinematic constraints introduce local reference frames.

Here, we study a specific example: `Reacher` (Easy and Hard). The second joint has angle $\theta_2$ and angular velocity $\dot{\theta}_2$ relative to the first link.

For *local* reference frame version, we use

$$\left(\theta_1, \theta_2, \dot{\theta}_1, \dot{\theta}_2, x_g - x_f, y_g - y_f\right) \Rightarrow \left(\cos\theta_1, \sin\theta_1, \cos\theta_2, \sin\theta_2, \dot{\theta}_1, \dot{\theta}_2, x_g - x_f, y_g - y_f\right) \tag{58}$$

Thus, $\cos\theta_1, \sin\theta_1$ is transformed under standard representation $\rho_1$ and $\cos\theta_2, \sin\theta_2$ is transformed under trivial representation $\rho_0 \oplus \rho_0$.

For the *global* reference frame version, we compute the global location of the end-effector (tip) by adding the location of the first joint. Thus, the global position is transformed also under standard representation now $\rho_1$.

We show the results in Figure 15. Evaluation reward curves for non-equivariant and equivariant TD-MPC over 5 runs using global frames. Error bars denote 95% confidence intervals. Non-equivariant TD-MPC outperforms equivariant TD-MPC. Surprisingly, we find using global reference frame where the second joint is associated with standard representation (equivariant feature, instead of invariant feature) brings much worse results, compared to the local frame version in the main paper. One possibility is that it is more important to encode kinematic constraints (e.g., the length of the second link is preserved in $\cos\theta_2, \sin\theta_2$), compared to using equivariant feature.

## G ADDITIONAL MATHEMATICAL BACKGROUND

We provide additional mathematical background on representation theory and its relation with our theoretical results.

### G.1 MATHEMATICAL EXPOSITION: PRINCIPAL $G$-BUNDLES AND LQR

Fiber bundles were introduced in the 1930s as a natural extension of the concept of tangent spaces in differential geometry (Milnor & Stasheff, 1974). Conceptually, fiber bundles attach additional information at each point on some underlying manifold. The connection between equivarient machine learning and the theory of fiber-bundles was first noted in (Cohen & Welling, 2016b; Weiler et al., 2018a). For continuous state-action manifolds, the dynamics can be viewed as some additional structure that depends on the state action manifold location. The results presented in the main text can be understood within the context of fiber bundle theory.

### G.1.1 Fiber Bundle

We briefly comment on how Equivarient LQR can be understood in terms of fiber bundles. For a full exposition on the theory of fiber-bundles, please see (KARSTOFT, 1992; Husemöller, 2013).

Formally, a smooth fiber bundle is specified by $(E, B, F, \pi)$ where $E$ and $B$ are smooth manifolds and $F$ is a vector space. $E$ is called the *total space* and $B$ is called the *base space*. The vector space $V$ is called the *fiber*. The map $\pi : E \to B$ is a smooth subjection called the projection map. The projection $\pi$ must satisfy a trivialization condition (Husemöller, 2013).

Sections are generalization of vector fields. A smooth section $s$ of a fiber bundle is a smooth map $s : B \to E$ such that

$$\forall x \in B, \quad \pi(s(x)) = x$$

so that a section does not change the base point. One can think of a section $s(x)$ as a vector with origin at point $x \in B$.

### G.1.2 Principal Bundle

Intuitively, a $G$-Principal Bundle is fiber bundle with an additional symmetry on the total space. Let $G$ be a Lie group. Then, a smooth $G$-Principal Bundle is smooth fiber bundle that has continuous $G$ action on the total space $G \times E \to E$ which preserves the fibers of $E$ so that

$$\forall x \in E, \ \forall g \in G, \quad \pi(g \cdot x) = \pi(x)$$

Sections of $G$-Principal Bundles are defined analogously to the vector bundle case. Specifically, a section $s$ is a map $s : B \to E$ such that

$$\forall x \in B, \quad \pi(s(x)) = x$$

Note that if $s : B \to E$ is a section then $g \cdot s : B \to E$ is another section as

$$\forall x \in B, \ \forall g \in G \quad \pi(g \cdot s(x)) = \pi(s(x)) = x$$

Thus, on $G$-Principal bundles we can speak of $G$-families of sections $s_g : G \times B \to E$ defined as $s_g(x) = g \cdot s(x)$. It is thus natural to consider equivalence classes of sections. These equivalence classes correspond to flows related by $G$ action.

More formally, the LQR system in consideration is a $G$-bundle with total space given by $E = \mathcal{S} \times \mathcal{A}$ and base space $B = \mathcal{B}$ given by $G$-orbit representatives. The canonical projection $\pi : \mathcal{S} \times \mathcal{A} \to B$ is the map unto $G$-orbit representatives which satisfies

$$\forall g \in G, \quad \Pi(g \cdot x) = \Pi(x)$$

The LQR dynamical matrices $A$ and $B$ can be viewed as sections of the $G$-bundle. By utilizing equivarience methods, we only need to learn a section on the base space $B$ as opposed to vector fields on the total space $E$. This corresponds to learning equivalence classes of sections, instead of individual sections.