# OpenReview forum: "Can Euclidean Symmetry Help in Reinforcement Learning and Planning?"
_ICLR.cc/2024/Conference — Submitted to ICLR 2024_

### Official Review · Reviewer_YD5X · 2023-10-20

**Soundness:** 3 good
**Presentation:** 1 poor
**Contribution:** 2 fair
**Rating:** 3
**Confidence:** 3

**Summary:**

- The authors study equivariant model-based reinforcement learning.
- They first show that E(n)-equivariant MDPs give rise to an equivariant Bellman operator.
- They then focus on methods based on linearization of the MDP and show that LQR can be made equivariant.
- Their main contribution is an equivariant version of MPPI based on TD-MPC.
- They implement a version of this algorithm for discrete subgroups of E(2) and E(3) and demonstrate it on toy experiments (moving a point mass) and simple robotic gripping tasks.

**Strengths:**

- E(3) equivariance in robotics is compelling: many environments have underlying symmetries and data is often expensive to generate.
- As far as I know, this work is the first to study how to make MPPI equivariant (though not the first to study equivariant model-based RL, see below).
- The evaluation on robotic reaching tasks is relevant.

**Weaknesses:**

- The paper does not adequately discuss two important aspects of equivariance in robotics: What if the equivariance is latent, i.e. the observations are not given in known, simple representations of the symmetry group – instead, they are given in pixels? And what if the symmetry group is partially broken, for instance by object positions or the direction of gravity?
- The main contribution – the equivariant modification of MPPI – leaves some questions open (see below), which I'm sure can be addressed during the rebuttal.
- The paper presents the discussion of continuous groups as a main contribution. But it does not actually talk about architectures for equivariance to continuous groups, and all the experiments stick to discrete subgroups.
- Most of the experiments are on toy settings where the benefits of equivariance is obvious. It would be more interesting to see if these benefits carry over to more complex environments.
- The authors miss some references on equivariance in model-based RL and planning:
	- A. Deac at al, "Equivariant MuZero", arXiv:2302.04798
	- J. Brehmer et al, "EDGI: Equivariant Diffusion for Planning with Embodied Agents", arXiv:2303.12410
- The paper writing could be improved. Theoretical results like Theorem 2 should be stated more precisely in the main paper. In Sections 3 and 4, I found it difficult to follow the flow of the arguments. (The problem may not be the paper, though, but come from my lack of in-depth knowledge of MPC.)

**Questions:**

- As the most minor nitpick of all, why do you cite Einstein's special relativity paper for E(3) equivariance? That was the one work that made clear that E(3) is *not* the fundamental symmetry group of nature ;)
- In Section 4, the proposed method require a "G-steerable equivariant MLP" for continuous groups G. What architectures do you have in mind? If I'm not mistaken, you never experiment with any such architecture for continuous G, right?
- To make MPC equivariant, the action sampling needs to be equivariant. Just after Eq. (11), you describe as the main problem that "action sampling is not state-dependent". Why? Isn't the equivariant learned policy used for action sampling?
- I wasn't able to follow the logic behind the G-sample method, could you explain that a bit more slowly, please?
- Is it fair to compare non-equivariant and equivariant methods with equal parameter count? Usually, equi methods have less parameters than non-equivariant counterparts with similar capabilities. This choice thus strikes me as a bit unfair to the baselines.

---

> ### Author Response · Authors · 2023-11-22
>
> Thank you for your thorough review and insightful comments. We have addressed each of your concerns as follows:
>
> 1. **Equivariance in Robotics – Latent and Partial Symmetry Breaking:**
>
>     Your question about latent equivariance and partial symmetry breaking in robotics is indeed a fascinating area of research. In our study, we focused on scenarios where symmetry is known and explicit, as this forms the foundational basis for our theoretical exploration. While integrating latent equivariance or dealing with partially broken symmetry groups, such as those influenced by object positions or gravity, is highly relevant, we considered it beyond the scope of our current research. This is partly because (1) understanding the usefulness of symmetry in known cases is a preliminary step before delving into more complex scenarios, and (2) the problem of encoding unknown / latent / partially broken symmetries has been addressed in other works, such as Park et al., ICML 2022. However, we acknowledge that this is an important direction for future research and intend to explore it in subsequent studies.
>
> 2. **G-steerable Equivariant MLP for Continuous Groups + Contribution on Continuous Groups:**
>
>     Linearization of G-steerable equivariance constraints requires continuous G. We use the theory to show that it’s useful to have continuous group in continuous action space. The sampling-based algorithm is designed for continuous action, and can use discrete groups to still benefit sampling-based algorithm.
>
>     In short, continuous group + continuous action is required to derive theory, while we can use discrete group for sampling-based algorithm for continuous action space.
>
> 3. **Paper Citations:**
>     - The two mentioned papers are indeed relevant but also quite new. We will discuss them in the related work.
>     - The citation of Einstein's paper was a historical reference. We will ensure its relevance and appropriateness in our revised manuscript.
> 4. **Flow of Arguments and Precision of Theoretical Results:**
>     - We are revising sections with theoretical results, like Theorem 2, for improved precision and clarity. Sections 3 and 4 are being reorganized to enhance the flow and coherence of arguments.
> 5. **MPC Equivariance and Action Sampling:**
>
>     In our framework, there are two main components to consider in the context of action sampling:
>
>     1. **Policy Network:**
>         - The policy network plays a significant role in our approach, contributing approximately 25 samples in each Cross-Entropy Method (CEM) iteration (based on 512 total samples with about 5% coming from the policy network). This policy network is crucial for guiding the action sampling process towards promising regions of the action space.
>     2. **Sampling from a Fitted Gaussian Distribution:**
>         - In addition to the policy network, we also sample actions from a Gaussian distribution fitted to the top-performing samples. Each CEM iteration involves sampling 512 actions from this distribution. The purpose of CEM here is to iteratively adjust the Gaussian distribution (mean and variance) towards an optimal set of actions for the given state.
>
>     However, a key point to note is the initialization of the Gaussian distribution. For each sampling process, we initialize the mean and variance from a standard normal distribution, N(0,1)*N*(0,1), and iterate this process six times. This initialization approach means that the mean and variance of the Gaussian distribution are not directly dependent on the current state of the system.
>
>     The rationale behind this choice is to maintain a balance between exploration and exploitation. By not making the Gaussian parameters directly state-dependent, we allow for a degree of exploration which is critical in scenarios where the optimal action is not immediately apparent from the current state alone.

---

> > ### Author Response · Authors · 2023-11-22
> > **Continued**
> >
> > 6. **Fairness in Comparing Equivariant and Non-Equivariant Methods:**
> >     1. **Equal Parameter Count in Our Comparative Analysis:**
> >         - In our study, we have intentionally used an equal number of free parameters for both equivariant and non-equivariant methods. The primary difference lies in the structures and inductive biases of the architectures rather than the quantity of parameters. This approach follows a widely accepted convention in the field for comparing different equivariant neural networks, allowing for a more direct assessment of the impact of the architectural differences on performance.
> >     2. **Equivariance in Common Architectures:**
> >         - We would also like to point out that while certain architectures are typically referred to as non-equivariant, many common architectures inherently possess equivariant properties. A notable example is the Convolutional Neural Network (CNN), which exhibits translation equivariance. Thus, the distinction between equivariant and non-equivariant methods can be more nuanced than it initially appears, as many standard architectures integrate forms of equivariance in their design.
> >     3. **Focus on Architectural Differences:**
> >         - Our comparison aims to highlight how the specific structural features and inductive biases of equivariant architectures contribute to their performance, especially in contexts where symmetry properties play a crucial role. By maintaining an equal parameter count, we strive to ensure that any observed performance differences are attributable more to these architectural aspects than to the sheer number of parameters.
> >
> >     We believe this approach provides valuable insights into the efficacy of equivariant architectures and contributes to a more nuanced understanding of their role in various applications.
> >
> > 7. **Logic Behind the G-sample Method:**
> >     1. **Equivariance in the G-sample Method:**
> >         - In our Model Predictive Path Integral (MPPI) framework, we sample $N$ actions from a Gaussian distribution  $N(\mu, \sigma^2I)$, creating a set $A = \{a_i\}^N_{i=1}$ . A key aspect of our method is ensuring that the action sampling process maintains equivariance.
> >         - Equivariance necessitates that any transformation of the input state (for instance, a rotation) should correspond to a similar transformation in the output. However, since the parameters $\mu$ and $\sigma$ of the Gaussian distribution in standard MPPI are not state-dependent, they do not naturally satisfy this condition.
> >     2. **Augmenting Action Sampling for Equivariance:**
> >         - To address this, we augment the action sampling by transforming it with all elements in the group $G$, forming a new set $GA = \{g \cdot a_i | g \in G\}^N_{i=1}$.
> >         - This augmentation ensures that our method can handle various orientations, maintaining the property of equivariance. For each action $a_i$ in the original set $A$, the augmented set $GA$ includes the transformed action $g \cdot a_i$ for every element $g$ in $G$.
> >     3. **Invariance and Closure Under Group $G$:**
> >         - The invariance property we aim to achieve is that the return of a trajectory, $\text{return}(\tau)$, remains unchanged under transformations in $G$. This means the trajectory's return is invariant to rotations or other symmetries defined by $G$.
> >         - By augmenting the action set $A$ with transformations from G, we ensure that the set of potential actions (the sample space) is closed under $G$. This closure means that applying any transformation from $G$ to an action in our set results in another action that is also in our augmented set $GA$.
> >
> >     This methodical approach is crucial for maintaining consistency across different transformations and is a key aspect of achieving both equivariance and invariance in our planning algorithm.
> >
> >     We have detailed the mathematical foundations and further explanation of this process in Appendix E of our paper. We hope this comprehensive response clarifies the G-sample method's logic, implementation, and its critical role in our research.
> >
> >
> > We appreciate the opportunity to refine our manuscript based on your feedback. Your insights have been invaluable in guiding our efforts to enhance the quality and impact of our work.

---

> > ### Comment · Reviewer_YD5X · 2023-11-22
> >
> > I'd like to thank the authors for the detailed response. I understand some aspects of the work better now.
> >
> > Overall, I think this work could become a strong paper, but I am also convinced that it needs another iteration. I would politely suggest to motivate the design choices in the equivariant MPC better, add models that exhibit the full symmetry group and not just a discrete subgroup, run experiments on more interesting environments, and / or streamline the writing. In its current form, even with promises of improvements for the camera-ready version, I cannot recommend acceptance yet.

---

### Official Review · Reviewer_9qqP · 2023-10-31

**Soundness:** 1 poor
**Presentation:** 2 fair
**Contribution:** 1 poor
**Rating:** 3
**Confidence:** 5

**Summary:**

This paper studies continuous symmetry in model-based planning by showing that such MDPs have linear approximations that satisfy steerable kernel constraints. The proposed algorithm follows an MPC style algorithm and is evaluated on a few tasks with continuous symmetries.

**Strengths:**

1. The paper is well-motivated and well-written.
2. The proposed methodology is accompanied with theoretical analysis for giving insight into the use of symmetry in model-based planning.

**Weaknesses:**

### 1. Experimental setup and the baselines
* The choice of the baselines is very limited. The proposed algorithm is only compared against a non-equivariant TD-MPC. I would consider the other baselines as ablation studies of the proposed algorithm with different subgroups ($D_8$, $D_4$, $C_8$). To better evaluate the performance of the algorithm, I suggest the authors use at least one other baseline from the literature on symmetry in continuous control RL, such as [1], and at least another baseline from model-based RL, such as Dreamer [2].

* The proposed algorithm is evaluated on only four environments, two of which (2D point mass and 3D point mass) are toy problems. In fact, the 2D point mass is mainly used for debugging purposes and is rarely reported in a scientific paper due to its simplicity. I suggest the authors incorporate more experiments either from the robotics literature or works on continuous symmetry in RL, such as [3].

* Finally, I strongly encourage the authors to look into [9, 10] and follow their guidelines for reporting statistically significant results in RL.

### 2. Overclaims and missing related work
* The authors have overlooked some essential papers, and I have identified some of their stated contributions as overstated. On page 2 (first paragraph), they state that their approach expands on earlier research on planning on 2D grids, yet [4] has already examined the equivariant Muzero which integrates symmetries in a complex model-based planning algorithm. Furthermore, they assert that they are extending equivariant model-free RL to continuous states and actions, while [1] has already accomplished this in a broader context, not solely restricted to Euclidean symmetry. Some examples of other missing references are [5, 6].

### 3. Incremental contributions
* The definition of Geometric MDP appears to be a rebranding of MDP homomorphisms [7], which was also extended to continuous states and actions [1]. It is not clear why the authors have chosen to rename a well-studied concept in the literature by adding some restrictions on the group symmetry. This can be very misleading to an inexperienced reader.

* Additionally, the contributions of this paper appear to be incremental with respect to the prior work of [7] which explored the use of symmetry in model-based planning.

### 4. Discrepancy between theory and experiments
* One of the key contributions of the paper, as claimed by the authors, is the study of continuous group symmetries in RL. Unfortunately, in their practical algorithm they are using discretized subgroups (page 8). This raises doubts regarding the soundness of the paper and the connection between theoretical analysis and the experimental results.

### 4. Limiting assumptions
* The MDP dynamics is assumed to be deterministic (page 5, second paragraph) without any justification or insight into its reason.

### References

[1] Rezaei-Shoshtari, S., Zhao, R., Panangaden, P., Meger, D., & Precup, D. (2022). Continuous MDP Homomorphisms and Homomorphic Policy Gradient. Advances in Neural Information Processing Systems, 35, 20189-20204.

[2] Hafner, D., Lillicrap, T., Ba, J., & Norouzi, M. (2019, September). Dream to Control: Learning Behaviors by Latent Imagination. In International Conference on Learning Representations.

[3] Panangaden, P., Rezaei-Shoshtari, S., Zhao, R., Meger, D., & Precup, D. (2023). Policy Gradient Methods in the Presence of Symmetries and State Abstractions.

[4] Deac, A., Weber, T., & Papamakarios, G. (2023). Equivariant MuZero.

[5] Biza, O., & Platt, R. (2019). Online abstraction with MDP homomorphisms for Deep Learning. In Proceedings of the 18th International Conference on Autonomous Agents and MultiAgent Systems.

[6] Mahajan, A., & Tulabandhula, T. (2017). Symmetry learning for function approximation in reinforcement learning.

[7] Ravindran, B. (2004). An algebraic approach to abstraction in reinforcement learning. University of Massachusetts Amherst.

[8] Zhao, L., Zhu, X., Kong, L., Walters, R., & Wong, L. L. (2022, September). Integrating Symmetry into Differentiable Planning with Steerable Convolutions. In The Eleventh International Conference on Learning Representations.

[9] Henderson, P., Islam, R., Bachman, P., Pineau, J., Precup, D., & Meger, D. (2018, April). Deep reinforcement learning that matters. In Proceedings of the AAAI conference on artificial intelligence (Vol. 32, No. 1).

[10] Agarwal, R., Schwarzer, M., Castro, P. S., Courville, A. C., & Bellemare, M. (2021). Deep reinforcement learning at the edge of the statistical precipice. Advances in neural information processing systems, 34, 29304-29320.

**Questions:**

1. What are the key distinguishing features of Geometric MDPs compared to MDP homomorphisms?
2. Why is the MDP dynamics assumed to be deterministic? Which part of the algorithm breaks in the case of stochastic dynamics?

---

> ### Author Response · Authors · 2023-11-22
>
> - Thank you for your insightful comments and queries regarding our manuscript. We appreciate the opportunity to clarify these points and have addressed each of your concerns as follows:
> - **Relation to MDP Homomorphism, Choice of Baselines, and Incremental Contributions:**
>     - **Clarification on MDP Homomorphism:**
>         - It appears there may be a misunderstanding regarding our work's connection to MDP homomorphism. While our research is indeed based on the concept of MDP homomorphism, we emphasize that the original MDP homomorphism works do not provide practical algorithms for leveraging symmetries. Our approach aims to expand the practical application of MDP homomorphism by focusing on a smaller, yet significant and practical subset of problems where MDP homomorphism or isomorphism is induced by symmetry in the MDP. This focus allows us to propose more practical algorithms without the need for NP-hard orbit searches and known dynamics, which are significant limitations in the traditional application of MDP homomorphism. In our work, we use equivariance to constrain solutions within the framework of homomorphism, achieving "value equivalence" in a more practical and applicable manner.
>     - **Quotient MDP M/G and Continuous Group Action:**
>         - As discussed in Appendix B.1, our methodology involves planning in a quotient MDP M/G, made possible only when G has a continuous group action on the MDP state/action space. This quotient map is central to our approach and differentiates our work from prior studies.
>     - **Differentiation from Prior Work [8] and Expansion of Discussion:**
>         - We acknowledge that our contributions extend beyond the work presented in [8], which focused on discrete symmetry. Our research introduces a theory for continuous symmetry and an algorithm tailored for sampling-based algorithms in continuous action spaces. We have expanded the discussion of symmetry in MDPs and MDP homomorphism in a way that provides both theoretical depth and practical application, addressing a gap in the existing literature.
>     - **Discussion of Relevant Work [1]:** We acknowledge the relevance of work [1] in the context of continuous MDP homomorphisms. While we are prepared to compare our approach against it, we note several key differences:
>         - **Focus on Policy Gradient:** Work [1] primarily focuses on policy gradient methods, which differ from our model-based RL approach. This distinction is crucial as it influences the fundamental methodologies and outcomes of the respective research.
>         - **Specialization in Equivariance:** Unlike our work, [1] does not specialize in the equivariance of the system and does not explicitly take the symmetry group as an input. This difference could result in an unfair comparison, as our work and [1] address different aspects of MDPs.
>         - **Comparison with Other Equivariant RL Works:** To our knowledge, [1] has not been commonly compared in other equivariant RL works, making it unclear how it fits within the broader context of equivariant systems. Despite these points, we are open to comparing our approach with [1] to provide a comprehensive analysis.

---

> ### Author Response · Authors · 2023-11-22
> **continued**
>
> - **Distinction of Geometric MDP:**
>
>     The uniqueness of Geometric MDP in our work lies in the specialized application of MDP homomorphism principles, tailored to leverage the inherent symmetries in model-based reinforcement learning. Unlike traditional MDP homomorphism works, which often lack practical algorithms for exploiting symmetries, our Geometric MDP framework is designed with practicality and applicability in mind. This approach allows us to transcend the theoretical limitations of traditional MDP homomorphisms and offer concrete, actionable methodologies.
>
>     Key aspects that distinguish Geometric MDPs in our work include:
>
>     1. **Practical Application of Symmetries:**
>         - Our focus is on the practical use of symmetries within MDPs. By narrowing our scope to specific, symmetry-based problems, we provide more practical algorithms that do not rely on NP-hard orbit searches or known dynamics. This practical orientation is essential for making the concept of MDP homomorphism more accessible and applicable in real-world reinforcement learning scenarios.
>     2. **Equivariance Constraints:**
>         - A cornerstone of our approach is the use of equivariance to impose constraints on solutions, facilitating "value equivalence" in a manner that is both theoretically sound and practically viable. This approach allows us to exploit symmetries without the need for exhaustive searches or a priori knowledge of the dynamics.
>     3. **Planning in Quotient MDP M/G:**
>         - Our methodology involves planning in a quotient MDP M/G, which is feasible only when there is a continuous group action on the MDP state/action space. This quotient map, discussed in detail in Appendix, is a unique feature of our approach that leverages continuous group actions to provide a more structured and symmetrical framework for decision-making.
>     4. **Focus on Continuous Symmetry:**
>         - In contrast to prior works that primarily focus on discrete symmetry, our research introduces and develops a theory for continuous symmetry. This theory is pivotal for algorithms designed for sampling-based methods in continuous action spaces, representing a significant advancement in the field.
>
>     In summary, Geometric MDPs in our work represent a novel approach to MDP homomorphism, tailored to the demands and challenges of modern reinforcement learning. By bridging the gap between theoretical concepts and practical application, we offer a framework that is both innovative and applicable, addressing the complexities of continuous control tasks and the need for efficient, symmetry-aware decision-making algorithms.
>
> - **Choosing baselines and tasks.**
>
>     In response to your suggestions regarding the expansion of our experimental setup, we are considering incorporating additional tasks from reference [3] into our study. This inclusion will allow us to demonstrate the applicability and efficacy of our approach in a broader range of environments, particularly those that emphasize continuous symmetry in reinforcement learning.
>
>     Regarding the inclusion of additional baselines, specifically Dreamer as cited in [2], we wish to clarify that our primary focus lies in demonstrating the usefulness of equivariance in model-based reinforcement learning, particularly in continuous control contexts. The approach we are based on, Temporal Difference Model Predictive Control (TDMPC), has been shown to outperform Dreamer in continuous control tasks. While we acknowledge the importance of comparing various model-based RL baselines, our objective is to highlight the specific benefits that equivariance brings to these algorithms, rather than to conduct a broad comparison across different model-based RL methods. This focused approach enables us to more effectively illustrate the unique advantages that our equivariant framework offers in handling continuous control challenges.
>
> - **Connection to equivariant MuZero.**
>
>     Regarding, equivariant MuZero, it is not just about “complex” model-based planning algorithm. MuZero is based on MCTS for action selection. We focus on continuous control and continuous symmetry.
>
> - **On using discrete symmetry.**
>
>     In short, continuous group + continuous action is required to derive theory, while we can use discrete group for sampling-based algorithm for continuous action space.
>
>     More concretely, linearization of G-steerable equivariance constraints requires continuous G. We use the theory to show that it’s useful to have continuous group in continuous action space. The sampling-based algorithm is designed for continuous action, and can use discrete groups to still benefit sampling-based algorithm.
>
> - **Stochastic dynamics.**
>
>     The sampling-based algorithms do not need determinisiticity. Only the theory assumes determinisiticity, while it can be easily extended to Linear Quadratic Gaussian and considers stochasticity.

---

> > ### Comment · Reviewer_9qqP · 2023-11-23
> >
> > I thank the authors for their reply. Overall, I think this could potentially be a strong paper, but unfortunately I was not convinced by the authors’ reply. The major issues remaining after the rebuttal are:
> > * Toy experimental environments from low-dimensional states as reviewer **YD5X** also stated.
> > * Overclaims and a disconnect from the literature on MDP homomorphisms: In their rebuttal, the authors have compared their method against the original MDP homomorphism work, whereas both its theory and practical algorithms have been extended significantly over the past 20 years.
> > * Lack of relevant baselines: It appears that the authors are not convinced about this limitation on comparing against other model-based and equivariant algorithms.
> > All of these require a significant change beyond a revision, therefore in its current state, I cannot recommend acceptance.

---

### Official Review · Reviewer_NZJg · 2023-10-31

**Soundness:** 3 good
**Presentation:** 3 good
**Contribution:** 3 good
**Rating:** 6
**Confidence:** 3

**Summary:**

This work focuses on reinforcement learning and planning tasks that have Euclidean group symmetry. Motivated by geometric graphs, this work defines Geometric MDPs as the class of MDPs that corresponds to the decision process in Euclidean space. To investigate if Euclidean symmetry can guarantee benefits in model-based RL, this work presents a theoretical framework that studies the linearized dynamics of geometric MDPs. The theoretical results show that the matrices in linearized dynamics are G-steerable kernels, which can be used as a solution that significantly reduces the number of parameters. Inspired by the theoretical results, this work proposes an equivariant sampling-based model-based RL algorithm for Geometric MDPs. Empirical results in DeepMind Control suite demonstrated the effectiveness of the proposed method with continuous symmetries.

**Strengths:**

1.	The paper is well-written and formatted. The presentation of this work well connects the prior work: (1) value-based planning, (2) model-free equivariant RL, and (3) geometric deep learning.

2.	The first contribution of this work is Geometric MDPs, which define a class of MDPs with geometric structure and extend a previously studied discrete case to a continuous case. The symmetry properties in the Geometric MDPs are specified by equivariance and invariance of the transition and reward functions respectively.

3.	The second contribution is providing theoretical guidance on assessing the potential benefits of symmetry in a Geometric MDP for RL. Focusing on linearized Geometric MDPs, the theory shows that the matrix-value function satisfies G-steerable kernel constraints, which is useful for parameter reduction. They also found that tasks have dominated global Euclidean symmetry and less local symmetry can have relatively better parameter reduction.

4.	Based on the theory, they extend previous work TD-MPC to incorporate symmetry into sampling-based planning algorithms. The implementation is performed to ensure several components satisfy G-equivariance.

**Weaknesses:**

1.	Although Euclidean symmetry can bring significant savings in parameters, it does not always offer practical benefits for some tasks with local coordinates, e.g., the locomotion tasks.

2.	The proposed method assumes the symmetry group is known, which may limit its practical application.

**Questions:**

1.	Since the proposed method extends previous work from discrete case to continuous case, can the proposed method also cover the tasks with discrete actions?

2. How is the performance of the proposed method compared to other well-known RL algorithms, e.g., SAC, and DDPG on DeepMind Control suite?

---

> ### Author Response · Authors · 2023-11-21
>
> Thank you for your insightful comments and queries regarding our manuscript. We appreciate the opportunity to clarify these points and have addressed each of your concerns as follows:
>
> 1. **Theoretical Foundation and Applicability to Other Tasks:**
>     - We agree with your observation regarding the theoretical basis of our work. Our intention is indeed to build a robust theoretical framework to understand when and how we can have Euclidean symmetry in model-based reinforcement learning. While our current focus is on specific tasks, the underlying theory has potential applications in a broader range of tasks, which we aim to explore in future research.
> 2. **Assuming Known Symmetry:**
>     - Our study primarily concentrates on equivariance between reference frames in D-dimensional spaces, focusing on distance-preserving transformations. Normally, the control tasks’ dynamics we are interested in are symmetric under these  distance-preserving transformations. The cornerstone of this approach is SO(D) equivariance, which we believe is crucial for understanding and leveraging symmetries in these contexts.
> 3. **Applicability to Tasks with Discrete Actions:**
>     - Technically, our approach can be extended to tasks with discrete actions. However, our current focus is on continuous control tasks where we utilize sampling-based algorithms like Cross-Entropy Method (CEM) and Model Predictive Path Integral (MPPI). These algorithms iteratively update a Gaussian distribution, which is well-suited for our study. We acknowledge that other parameterizations or methods like Monte Carlo Tree Search (MCTS) could be used, and this is an area we might explore in future work.
> 4. **Performance Comparison with Other RL Algorithms:**
>     - We understand the importance of a comprehensive performance comparison with other well-known RL algorithms. It’s worth to note that TDMPC already outperformed SAC in their original paper. We also included a “planning-free” equivariant TDMPC to mimic equivariant SAC, which was outperformed by the version with planning, as shown in Figure 11, “Ablation on model-based vs. model-free (“planning-free”)”.

---

### Official Review · Reviewer_7StE · 2023-11-01

**Soundness:** 2 fair
**Presentation:** 1 poor
**Contribution:** 2 fair
**Rating:** 5
**Confidence:** 3

**Summary:**

The paper presents a new algorithm for utilizing domain-inherent symmetry during training in RL. In contrast to previous work, it claims to first define a class of Geometric MDPs and provide practical implementations, which outperform a vanilla RL approach.

**Strengths:**

The paper tackles an interesting issue. Inherent symmetries are definitely under-utilized in current RL approaches in many ways. Incorporating the symmetry directly into the MDP (and not possibly other parts of the training loop) certainly warrants some additional exploration. Using a theoretical motivation to identify suitable domains is a good idea and presenting these domains should be (in an extended form) part of this paper's contribution.

The paper has been well proof-read for typos.

**Weaknesses:**

The main issue I have with this paper is that it reads like a primer for its own appendix, which is not how I reckon papers should be written. In general, the Appendix is referred to way too often (sometimes even sneakily like in a reference to Figure 11) and it thus contains way too many crucial parts of the overall argument. In contrast, the contribution is not as novel or groundbreaking that building up such a huge apparatus seems justified. A good theoretical class description and an extensive empirical study would have been appreciated.

As of now, the paper lacks focus on multiple fronts. various Theorems, Propositions, paragraph titles and types of lists make up the main body, but do not give it structure as the parts to not naturally follow from one another. As if the paper knows that it lost some people, the paper provides a recap on its own at the beginning of section 4. Perhaps beginning with the empirical study and the example domains and deriving the theoretical class from there would be easier?

The empirical study does a good job at motivating further research but lacks a definite conclusion. Most importantly, it would be nice to know how the described behavior translates to other means of training for MDPs.

Minor notes:
- "mappingto" instead of "mapping to" (p. 4)
- "demonstrat" instead of "demonstrate" (p. 5)

**Questions:**

see above

---

> ### Author Response · Authors · 2023-11-21
>
> Thank you for your thorough review and insightful feedback on our manuscript. We are actively addressing each of your concerns as follows:
>
> 1. **Concern about the Manuscript Overly Relying on the Appendix:**
> We are revising the manuscript to ensure that the main text is self-contained and robustly presents our key concepts and findings. We are reducing references to the appendix and ensuring that it serves only as a supplement. This approach will enhance the readability of our paper and more clearly showcase our novel contributions.
> 2. **Lack of Focus and Structure:**
> Your suggestion is helpful for our reorganization of the paper. In response to your feedback, we are actively considering reorganizing the paper to better motivate the paper with our empirical study.
> 3. **Empirical Study and Conclusions:**
> We are enhancing our empirical study to provide clearer and more definitive conclusions. Additionally, we are exploring how our findings could influence other training methods for MDPs, particularly in the realm of model-based reinforcement learning with continuous action and symmetry groups. We recognize that our specific focus might limit direct applicability to other training methods; however, we see potential in areas such as imitation learning when demonstrations are available.
>
>     Additionally, we restate our contributions from the empirical study:
>
>     1. **Demonstration of Euclidean Symmetry in Model-Based RL Algorithms:**
>         - The paper defines a subclass of Markov Decision Processes (MDPs), termed Geometric MDPs, that are prevalent in robotics and exhibit additional structural properties. These MDPs, when linearized, adhere to steerable kernel constraints, leading to a substantial reduction in the parameter space.
>     2. **Development of a Novel Model-Based RL Algorithm:**
>         - Utilizing the insights gained from the theoretical framework, the paper develops a model-based RL algorithm that leverages Euclidean symmetry. This algorithm is shown to outperform standard techniques in common RL benchmarks.
>     3. **Consideration of Equivariance in Sampling-Based RL Methods:**
>         - The paper claims to be the first to consider the importance of equivariance in sampling-based RL methods. This contribution provides a deeper understanding of symmetry in RL algorithms and offers insights for future research.
>     4. **Practical Limitations and Scope:**
>         - While Euclidean symmetry can lead to significant savings in parameters, it does not always offer practical benefits for tasks with local coordinates. For example, locomotion tasks do not greatly benefit from this approach.
>         - The algorithm assumes that the symmetry group is known, which is typically determined by the robot's workspace dimension (often 2D or 3D).
>     5. **Implications and Future Directions:**
>         - The findings underscore the value of Euclidean symmetry in model-based RL algorithms, especially in robotics. However, the paper also acknowledges that further research is needed to explore the full potential and limitations of this approach, particularly in tasks with different characteristics or constraints.
>
> We appreciate your constructive feedback, which has been instrumental in guiding our revisions.

---

### Meta-Review · Area_Chair_RZY9 · 2023-12-09

**Metareview:**

This paper studies a very interesting and important problem in equivariant representation learning and reinforcement learning. I believe this is an under explored topic in the research community and I was excited to see it being explored by the authors. This paper focuses on equivariant model-based RL, with a focus on integrating continuous symmetry into model-based planning. It showcases the development of equivariant MDPs that lead to an equivariant Bellman operator and extends this concept to demonstrate how Linear Quadratic Regulator (LQR) can be adapted for equivariance. The paper's primary innovation is the creation of an equivariant version of the Model Predictive Path Integral (MPPI) based on Time-Delayed Model Predictive Control (TD-MPC). This method is demonstrated on simple experiments -- moving a point mass and robotic gripping tasks.

Key strengths of this research include its pioneering work in making MPPI equivariant. This is especially relevant to the field of robotics, considering the prevalent symmetries in such environments and the high cost of data generation.

However, I believe the paper requires more work and should incorporate all the feedback received from this review cycle to make it stronger. There were several questions and limitations that need to be addressed. For instance, the paper does not sufficiently address latent equivariance or scenarios where the symmetry group is partially broken. These issues are critical in more complex environments where observations are not presented in simple, known representations of the symmetry group. As pointed out by the reviewers, the authors should more effectively explain the rationale behind their design choices in the equivariant MPC, incorporate models that demonstrate the entire symmetry group rather than just a discrete subgroup, and conduct experiments in more diverse and complex settings. Moreover, most experiments are limited to toy settings, and there's a lack of exploration in more complex environments to demonstrate the benefits of equivariance.

I believe the paper needs further rounds of revision and broader experimentation before it is ready for publication. I think this research area is a very interesting topic for ICLR and beyond I want to encourage the authors to resubmit this paper after a revision.

**Justification For Why Not Higher Score:**

N/A

**Justification For Why Not Lower Score:**

N/A

---

### Decision · Program_Chairs · 2024-01-16

Reject